# Tyrosine 121 moves revealing a ligandable pocket that couples catalysis to ATP-binding in serine racemase

Chloe R. Koulouris[1], Sian E. Gardiner [2], Tessa K. Harris[2], Karen T. Elvers[2], S. Mark Roe [3], Jason A. Gillespie[2], Simon E. Ward [2], Olivera Grubisha[2], Robert A. Nicholls [4], John R. Atack [2✉] & Benjamin D. Bax [2✉]

Human serine racemase (hSR) catalyses racemisation of L-serine to D-serine, the latter of which is a co-agonist of the NMDA subtype of glutamate receptors that are important in synaptic plasticity, learning and memory. In a 'closed' hSR structure containing the allosteric activator ATP, the inhibitor malonate is enclosed between the large and small domains while ATP is distal to the active site, residing at the dimer interface with the Tyr121 hydroxyl group contacting the α-phosphate of ATP. In contrast, in 'open' hSR structures, Tyr121 sits in the core of the small domain with its hydroxyl contacting the key catalytic residue Ser84. The ability to regulate SR activity by flipping Tyr121 from the core of the small domain to the dimer interface appears to have evolved in animals with a CNS. Multiple X-ray crystallographic enzyme-fragment structures show Tyr121 flipped out of its pocket in the core of the small domain. Data suggest that this ligandable pocket could be targeted by molecules that inhibit enzyme activity.

[1] Sussex Drug Discovery Centre, University of Sussex, Brighton BN1 9QG, UK. [2] Medicines Discovery Institute, Cardiff University, Cardiff CF10 3AT, UK. [3] Department of Biochemistry and Biomedicine, University of Sussex, Brighton BN1 9QJ, UK. [4] MRC Laboratory of Molecular Biology, Francis Crick Ave, CB2 0QH Cambridge, UK. ✉email: AtackJ@cardiff.ac.uk; baxb@cardiff.ac.uk

N-methyl-D-aspartate (NMDA) receptors (NMDARs) are a subtype of ionotropic glutamate receptor that mediate a $Ca^{2+}$-permeable component of excitatory neuro-transmission and are essential in neuronal processes such as synapse formation, synaptic plasticity, learning, and memory potentiation[1]. Overactivation of NMDARs results in a prolonged influx of calcium into neurons, which is reported to be cytotoxic[2–4]. Modulation of NMDAR activity is a viable strategy for treating various CNS disorders, as evidenced by the recent FDA approval of the NMDAR channel blocker esketamine for treatment-resistant depression[5,6]. In general, NMDARs are het-erotetrameric assemblies of an obligate GluN1 subunit and a variable GluN2 subunit arranged as a pair of heterodimers and for activation require binding of two agonists: glutamate and D-serine or glycine[1]. D-Serine and glycine bind to the same site in GluN1 but have distinct neuroanatomical distributions, with D-serine predominantly localised to the forebrain and glycine to the hindbrain and brainstem[7]. There is considerable evidence that D-serine is the more physiologically relevant neurotransmitter for modulating brain activity that underpins learning and memory in the forebrain[7–9]. D-Serine is involved in normal NMDAR-mediated neurotransmission, however aberrant regulation of intercellular D-serine is linked to the overactivation of NMDARs that occurs in neurodegenerative conditions[10]. Moreover, at rat brainstem hypoglossal motor neurons, for which glycine is the primary agonist, D-serine has the ability to elicit more potent activation at NMDARs even at concentrations 100-fold lower than glycine[11].

In mammals, D-serine is synthesised by a pyridoxal-5′-phosphate (PLP)-containing enzyme serine racemase (SR). In addition to the racemisation of D- and L-serine (Fig. 1a–c), SR can also act as a dehydratase[12], catalyzing the beta-elimination of the elements of water from L-serine, thereby producing ammonia and pyruvate, following enamine tautomerization and hydrolysis[13–15] (Fig. 1d, e). In PLP-dependent enzymes, such as SR, the PLP is initially linked to a lysine residue via a Schiff-base[16] or 'internal aldimine' (Supplementary Fig. 1). Upon substrate binding, the amine of the substrate can displace the lysine to form an 'external aldimine' (Fig. 1c, Supplementary Fig. 1). PLP can act as an electron sink, with monoanionic, dianionic and trianionic forms all being stable (Supplementary Fig. 1). PLP undergoes conformational changes during the cat-alytic cycle and numerous conformations have been observed[14]. There are many PLP-dependent enzymes that catalyse many different reactions and they are grouped into seven structural classes[16]. SR is a type II PLP-dependent enzyme and other members of the family include the β-subunit of tryptophan synthase[17], aspartate racemase[18], threonine deaminases and serine dehydratase (SDH)[19].

Mouse knock-out models of SR have brain D-serine levels 10–20% of wild-type (WT) controls[10] and also demonstrated schizophrenia-like symptoms that were ameliorated by D-serine[20]. This is consistent with studies showing that interaction of mutated Disrupted-In-Schizophrenia-1 (DISC1) protein with SR decreases D-serine levels and produces schizophrenia-like symptoms in mice[21]. Both SR inhibitors and activators have therapeutic potential as indirect modulators of NMDAR function in disorders associated with glutamate hyperfunction or hypofunction. For example, reducing D-serine release from inflammatory astrocytes may have therapeutic benefit in Alzheimer's disease by reducing NMDAR-mediated neurotoxicity[22]. Further understanding of how SR is activated or inhibited in vivo by interacting with proteins and small

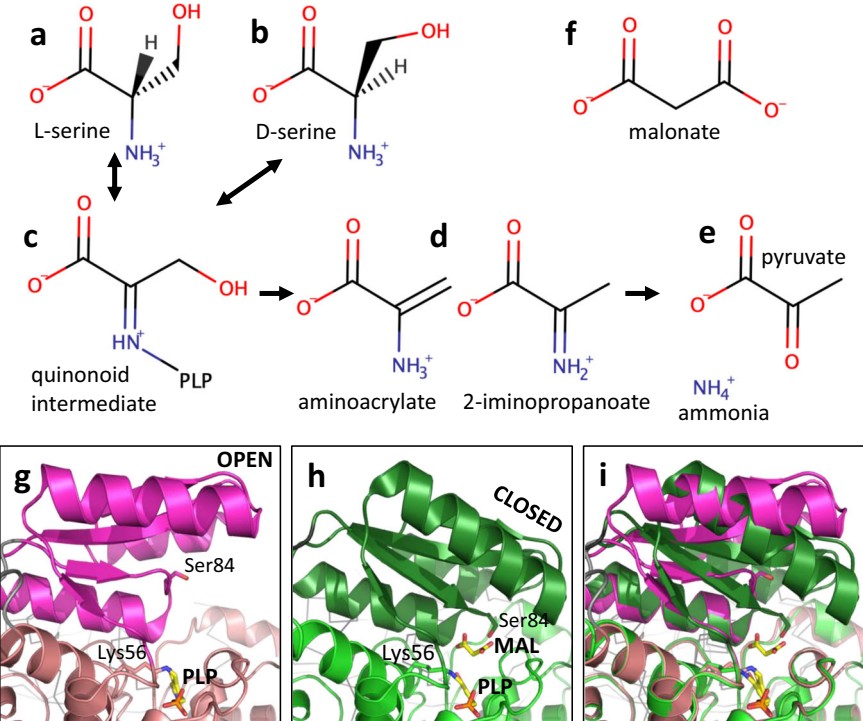

**Fig. 1 Overview of serine racemase. a–e** Simplified view of substrates and products in reactions catalyzed by serine racemase. Note the products, aminoacrylate and 2-iminopropanoate, are two interchangeable tautomers that will be deaminated to give pyruvate (and ammonia) in aqueous solution. **f** The substrate analogue inhibitor malonate. Chemical structures were drawn using MarvinSketch. **g** The 'open' 1.89 Å *holo* structure of hSR (pdb code: 6SLH). The structure is shown as a cartoon with stick representations of PLP (yellow carbons) and side-chains of Lys56 (pale pink carbons) and Ser 84 (magenta carbons). The small domain (78–155) is in magenta with the interdomain linker (69–77) in grey. The second subunit in the dimer is partially visible in the background (dark grey Cα trace). **h** The 'closed' 1.5 Å structure of hSR with PLP and malonate (pdb code: 3L6B). Side-chains of Lys 56 and Ser 84 are shown as sticks (green carbons) as are PLP and malonate (yellow carbons). **i** Superposition of (**g**) and (**h**) on large domain.

molecules[21,23–30] may facilitate drug discovery of this target for which no therapeutic interventions currently exist.

The first mammalian structures of SR, published in 2010[31], revealed a dimeric structure with each subunit consisting of a small domain (residues 78–153) and a large domain (residues 4–67 and 156–316). SR has been observed in an 'open' conformation[31–33] with the active site exposed and accessible in a cleft between the two domains. A 'closed' conformation of SR is observed with malonate (Fig. 1f, h, Supplementary Fig. 2b)[31]. Malonate is a substrate analogue inhibitor[34] with a reported inhibition constant (Ki) of 33 μM for human SR (hSR)[35]. Conformational changes that occur within the small domain when hSR structures are compared[33] place the side-chain hydroxyl of Ser 84, a key catalytic residue required to generate D-serine, close to the central carbon of malonate (Fig. 1h)[31]. Mutation of Ser 84 to an alanine blocks hSR racemisation but not L-serine elimination activity, consistent with the presence of an alanine at this position in human SDH[19]. The racemisation activity is also eliminated in S84D and S84N mutants[36], but all mutants retain β-elimination activity. Crystallography has trapped several intermediates of L-serine dehydration in a cystathionine γ-lyase, including an external aldimine structure with aminoacrylate and a 2-iminopropanoate (Fig. 1d) intermediate[14]. Another structure of a PLP-dependent enzyme with a gem-diamine intermediate of D-serine (Supplementary Fig. 3) was recently published in the PDB (pdb code: 6TI4).

In 2009, four crystal structures were reported of yeast (S. pombe) SR (2ZPU, 1V71, 1WTC, 2ZR8). S. pombe SR (spSR) has 38% sequence identity with hSR and a similar dimer interface[37,38]. In two of the spSR structures (2ZPU, 2ZR8) the cofactor PLP was unexpectedly linked to a modified lysinoalanyl (Supplementary Fig. 3) residue[38] rather than directly to the catalytic lysine[37,38]. At the end of a standard cycle, the catalytic lysine replaces the amino acid and regenerates the internal aldimine, allowing the enzyme to open and release a product (i.e., D-serine, L-serine, or aminoacrylate/2-iminopropanoate, which are converted in solution to pyruvate and ammonia). The two yeast structures with the lysino-D-alanyl residue are in closed conformations (Supplementary Fig. 2e), very similar to the closed conformations observed with mammalian SR and malonate (Supplementary Fig. 2b). Interestingly, although the other two yeast structures (1V71 and 1WTC) are in 'open' conformations, they differ significantly from hSR 'open' structures (Supplementary Fig. 2a, d, g). Holoenzyme hSR structures (5X2L, 6SLH) share a common 'open' (inactive) conformation for the small domain[33,39] that is not observed in the spSR holoenzyme structures (1V71 and 1WTC)[37]. In one of the 'open' yeast structures (1WTC) the ATP analogue AMPPCP was soaked into holo crystals and was found to bind at the dimer interface. Comparison of hSR and spSR holo structures reveals the binding pocket of for ATP in spSR does not exist in the 'open' hSR structures (Supplementary Fig. 2g).

Both the racemisation and elimination reactions of SR are activated to some degree by ATP, but not identically in mammals and yeast. The activating effect of ATP can be potentiated by the divalent cations $Mg^{2+}$, $Mn^{2+}$, and $Ca^{2+}$ [12,40–42]. The cation-nucleotide complex (e.g., MgATP) forms a high-affinity ligand which increases D-serine production[43,44]. Studies on human[43,45], rat, and yeast[37] SR have reported addition of ATP and other activators caused a decrease in Km with the same Vmax for the racemisation reaction, indicating SR activation by ATP is allosteric.

While there is a clear biological rationale for developing hSR inhibitors to reduce NMDAR hyperactivity in certain CNS disorders, progress in this area has been slow. In 2006, Dixon et al. reported slow-binding hSR inhibitors from high-throughput screening of combinatorial libraries[46] containing large functional groups such as phenyl and imidazole rings. These peptide-like inhibitors were not very potent, with the best having a Ki of 320 μM. A review of SR inhibitors in 2011[47] listed three polar inhibitors (malonate, L-erythro-3-hydroxy-aspartate and L-aspartic acid β-hydroxamate) as more potent (Ki < 100 μM) than the best peptide-like inhibitor reported by Dixon[46]. Careful optimisation of malonate-based inhibitors[48] using the hSR structure with malonate[31] produced more polar compounds, none of which were much more potent than malonate (IC$_{50}$ 67 μM). A 2017 paper published an inhibitor of mouse SR, 13 J, that could suppress neuronal overactivation in vivo[39] and produced better IC$_{50}$ values (140 μM) than malonate (770 μM). In 2020, a novel plant derivative SR inhibitor, madecassoside (IC$_{50}$ 26 μM), was reported and described as a far better inhibitor than malonate (IC$_{50}$ 449 μM)[49]. IC$_{50}$ values are known to be dependent on conditions under which they are measured. The large differences in values of IC$_{50}$ reported for inhibition of mouse and human SR by malonate in the literature (770, 449, and 67 μM), are unlikely to be entirely due to the inherent instability of serine racemase.

In this paper we describe the first 'closed' crystal structure of hSR bound to ATP. In this structure, Tyr 121 has 'flipped' outwards from the core of the small domain, where it is situated in the holoenzyme structures, to form H-bonds with the α-phosphate of ATP at the dimer interface. We propose that Tyr 121 plays a key role in communicating between the active and ATP sites of hSR. We also describe five 'open' structures of SR bound to fragments identified from an XChem crystallographic fragment screen. Two compounds displace the sidechain of Tyr 121 into the 'out' position; one carboxylic acid-containing compound binds to a conformationally flexible region of the small domain; and two compounds bind at the dimer interface. We use the new binding information from the ATP- and fragment-bound structures to illuminate the catalytic mechanisms of SR that are not well understood and propose a novel strategy for developing inhibitors of SR that block enzyme activity by 'trapping' SR in a conformation that inhibits the ATP-induced conformational changes in SR required for substrate hydrolysis.

## Results

### A 1.8 Å 'open' hSR structure and a 1.6 Å 'closed' structure with ATP and malonate.
A new 1.80 Å hSR holoenzyme (6ZUJ; Fig. 2a) containing $Ca^{2+}$, rather than $Mn^{2+}$ or $Mg^{2+}$ in the metal binding site, was solved in preparation for an X-ray crystallographic fragment screening campaign (XChem)[50]. A crystal structure of ATP-bound hSR with the inhibitor malonate (6ZSP; Fig. 2b) was determined to a resolution of 1.60 Å in space group $P2_12_12_1$ (Supplementary Table 1). In this structure, the binding mode of malonate (Supplementary Fig. 4a) is virtually identical to that of an earlier hSR structure[31] with no ATP (3L6B; Supplementary Fig. 4b). Malonate resides in the substrate binding site with Ser 84 pointing towards the central carbon (Supplementary Fig. 4); in the transition state this carbon is planar (see quinonoid intermediate; Fig. 1, Supplementary Fig. 1). Two ATP molecules sit at the dimer interface (Fig. 2b) but in the 1.8 Å 'open' structure (6ZUJ) the pocket for ATP does not exist (Supplementary Fig. 5; Supplementary Movies S1 and S2) and the conformation is very similar to that observed in other hSR holoenzyme structures such as 5X2L[32], 6SLH[33], and the A-subunit of rat SR in 3HMK[31].

The binding site for ATP is in a pocket between the two subunits (Fig. 2b). The two ATP-binding pockets (Fig. 2b, Supplementary Fig. 5) are related by the twofold axis of the dimer and are virtually identical; one is described here (for the other, reverse A and B subunits names). The adenine ring sits between Ala A117 (from the α5 helix in the small domain of the A-

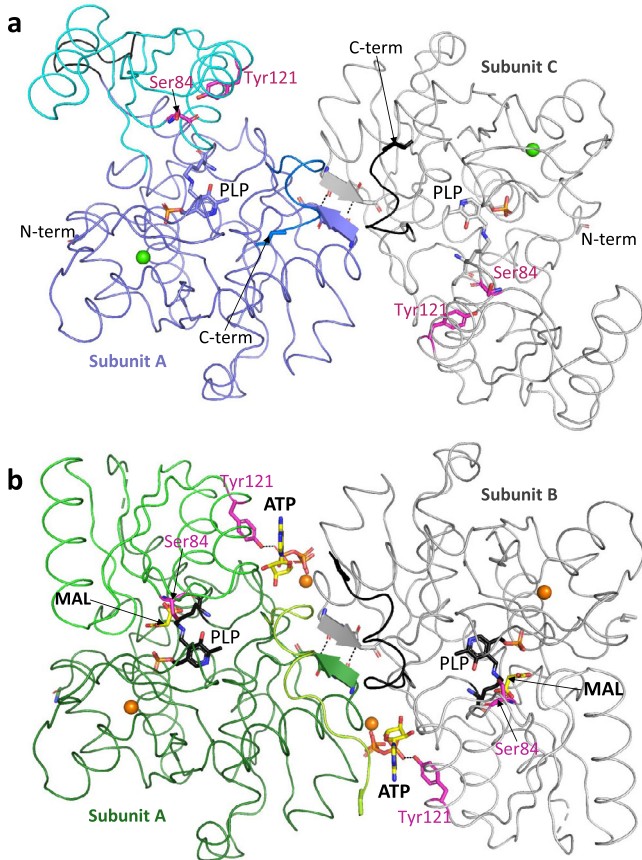

**Fig. 2 Crystal structures of 'open' and 'closed' human SR dimers. a** View of 1.8 Å 'open' *holo* SR structure (pdb code: 6ZUJ) with Ca$^{2+}$ ions (green spheres). The AC dimer is shown as a Cα ribbon with the A subunit in purple/cyan and the C subunit in grey. Also shown are Ser 84 and the side-chain of Tyr 121 (sticks, magenta carbons) and the PLP moiety covalently linked to Lys 56 (sticks, blue carbons). The hydroxyl of Tyr 121 is within H-bonding distance of the main-chain carbonyl oxygen of Ser 84 (dotted line) in subunit A. Residues 279–281 at the dimer interface form a short β-strand with two intersubunit H-bonds (dotted lines). **b** View down twofold axis of 1.60 Å human SR dimer (pdb code: 6ZSP) with ATP and malonate (sticks, yellow carbons) and Mg$^{2+}$ ions (orange spheres). The A/B dimer is shown as a Cα ribbon with the A subunit in green/dark green and the B subunit in grey. Also shown are Ser 84 and the side-chain of Tyr121 (sticks, magenta carbons) and the PLP moiety covalently linked to Lys 56 (sticks, black carbons). The side-chain hydroxyl of Tyr121 donates an H-bond to ATP (dotted line) in both A and B subunits.

subunit) and Arg B277 from the large domain of the B-subunit. Residues from both the large and small domains of the A-subunit make a number of hydrogen bonds to oxygens on the ribose and the α-phosphate of ATP (Supplementary Fig. 5a). Of note, the side-chain hydroxyl of Tyr A121 donates a hydrogen bond to an oxygen from the α-phosphate of ATP. The β and γ phosphates of the ATP interact with the B-subunit, accepting hydrogen bonds from residues at the N-terminus of the α2 helix (main-chain NHs from Ser B32 and Ile B33, and sidechain OHs from Ser B31 and Ser B32). A magnesium ion, coordinated by two oxygens from the β and γ phosphates of the ATP and four waters, does not directly contact the protein but helps orient the β and γ phosphates of the ATP, facilitating the interaction with the N-terminus of the α2-helix from subunit B.

The ATP-binding pocket does not exist in the 'open' holoenzyme structures because the position of the α5-helix (including A117) is very different and the two subunits in the dimer are not in the same relative positions in 'open' and 'closed' structures (Supplementary Movie S2).

**Magnesium and calcium ions in structures**. In the ATP-bound structure, Mg$^{2+}$ ions (Fig. 2b; orange spheres) are seen both adjacent to the ATP and in the metal-binding site (near the phosphate of PLP). SR activity is dependent on the presence of divalent metal ions. It has been reported that rat brain SR is activated in the presence of Mn$^{2+}$ ions (153% at 10 μM), Ca$^{2+}$ ions (134% at 1 mM) and Mg$^{2+}$ ions (112% at 10 μM)[12,42]. The original crystal structure of hSR with malonate, 3L6B[31] contains Mn$^{2+}$ ions, while our new structure with ATP and malonate (6ZSP) contains Mg$^{2+}$ ions, and our new 1.8 Å holoenzyme structure (6ZUJ) contains Ca$^{2+}$ in the metal binding centre (Fig. 2a; green spheres). In all hSR structures, the Mn$^{2+}$, Mg$^{2+}$ or Ca$^{2+}$ ions are similarly octahedrally coordinated by the main chain carbonyl of Ala 214, the side chains of Glu 210 and Asp 216, and three water molecules. Two of the three waters donate hydrogen bonds to backbone carbonyls where the corresponding main-chain NHs (from the tetra-glycine loop Gly 185–188 in hSR) donate H-bonds to the phosphate group of PLP[40], thus helping organise the PLP binding site. The Mg$^{2+}$ ion coordinated by the β and γ phosphates of ATP is also coordinated by four waters making up a standard octahedral coordination sphere. These Mg$^{2+}$ coordinating waters all donate hydrogen bonds to backbone carbonyls.

**Compounds can displace tyrosine 121 from the core of the small domain**. In published hSR holoenzyme structures (Supplementary Table 2), Tyr 121 sits in the core of the small domain[32,33]. We conducted an XChem crystallographic fragment screen using our new hSR *holo* crystal form, 6ZUJ (a representative 20% DMSO, no-compound control; Fig. 2a). Analysis of 600 fragment-soaked hSR crystals revealed 44 interesting binding events, of which twelve (Supplementary Table 3) had displaced Tyr 121 from the core of the small domain in at least one of the four subunits in the asymmetric unit. One of these is a 1.53 Å crystal structure of hSR bound to compound **1**-x0495 (Fig. 3a, A-subunit; Supplementary Table 4 for XChem ligand structures). The compound occupies a similar space to the sidechain of Tyr 121 in *holo* hSR structures as seen when compared with the 1.8 Å DMSO-soaked control structure (Fig. 3b, c).

A superposition of the 1-x0495 structure with the 1.6 Å malonate/ATP structure shows the Tyr 121 sidechains occupying similar positions in both structures (Fig. 3d). However, the structure with 1-x0495 bound remains in the 'open' conformation, with a cleft between the small and large domain that is accessible to substrate, whereas the structure with malonate and ATP is in the 'closed' conformation with Ser 84 in proximity to the malonate (Fig. 3d).

The ability to lock the enzyme in the 'open' inactive state appears to be important in higher eukaryotes. S-nitrosylation of Cys 113 turns SR 'off'[51–53]. In the 'open' conformation Cys 113 is readily accessible (Fig. 3a, b), however, in the complex with malonate and ATP the sulfur is buried in the hydrophobic core (Fig. 3d, Supplementary Fig. 6). Our structures suggest that S-nitrosylation of Cys 113 will stabilize the 'open' inactive conformation of the enzyme and is incompatible with the 'closed' conformation observed with malonate and ATP.

Many of the XChem fragments appear not to be very potent inhibitors of SR. Assay data for compounds observed to displace Tyr 121 (Supplementary Tables 3 and 4), showed only two reasonably potent inhibitors (3-x0458 and 4-x0482). The twelve XChem compounds binding to the Tyr 121 region exploit a naturally occurring pocket. Before the small domain can move to

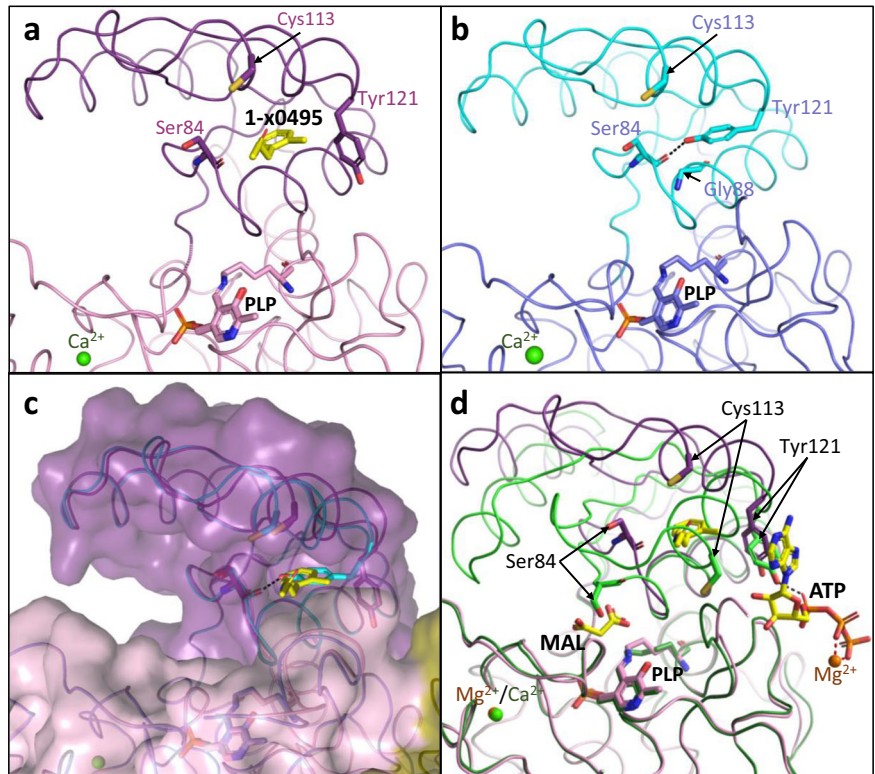

**Fig. 3 XChem compound 1-x0495 can displace tyrosine 121. a** View of subunit A from 1.53 Å human SR structure with compound **1**-x0495 from an XChem fragment screen (pdb code: 7NBG). Overall structure is shown as a Cα ribbon with stick representations of Ser 84 and the side-chains of Tyr 121 and Cys 113 (purple carbons). The PLP covalently attached to Lys 56 is also shown (sticks, pink carbons). **b** Equivalent view of 1.8 Å *holo*/ligand-free SR structure (pdb code: 6ZUJ) with Ca²⁺ ions (green sphere). The hydroxyl of Tyr 121 is within H-bonding distance of the main-chain carbonyl oxygen of Ser 84 (dotted line). **c** Superposition of the 1.53 Å structure with compound 1-x0495 (7NBG) with the 1.8 Å *holo*/ligand-free SR structure (6ZUJ). A semi-transparent surface has been drawn on 7NBG. **d** Large domain superposition of the 1.53 Å structure with compound 1-x0495 (7NBG; magenta ribbon) on the 1.6 Å malonate/ATP structure (6ZSP; green ribbon). Note Tyr 121 is in a similar position in both structures but Ser 84 and Cys 113 have moved.

enclose the substrate (or malonate), Tyr 121 needs to be expelled from the core of the small domain so that it can adopt the 'closed' catalytic hSR conformation (Supplementary Movie S2).

**Tyrosine 121 is in the core of the small domain in 'open' inactive hSR structures.** The small domain of hSR (residues 78–155) displays remarkable conformational flexibility between bound and unbound states[33] (Supplementary Movie S3, Supplementary Table 5). Catalysis occurs when the small domain moves to enclose the substrate L-Serine[31,33]. In the new 1.8 Å 'open' hSR structure (6ZUJ; Fig. 2a), Tyr 121 is largely buried in the core of the small domain (Fig. 3b, Supplementary Fig. 5). This buried position of Tyr 121 has been observed in two other *holo* hSR structures (5X2L, 6SLH) and in the A-subunit of the *holo* rSR (3HMK) structure[31–33]. There is a radical change in conformation of the side-chain of Tyr 121 between the malonate/ATP complex and the *holo* structure, with the side-chain hydroxyl group 'moving' approximately 10 Å from interacting with the α-phosphate of ATP (Fig. 2b) to the main-chain carbonyl of Ser 84 (Fig. 2a). This results in a major reorganisation of the structure of the small domain (Supplementary Fig. 5). Helix α4 from the small domain packs against the large domain and does not move significantly between the two structures, but the movement of Tyr 121 shifts helix α5 and the rest of the small domain (Supplementary Fig. 2) into a different conformation[33]. This positional shift of the tyrosine residue is not seen when yeast apo and ATP-bound structures are compared (Supplementary Fig. 2).

The different 'open' structure observed for yeast spSR seems to be due to a single residue change. In the human structures glycine

88, from the sequence SGNH**G**Q (84–89 in hSR) sits under Tyr 121 (Fig. 3b). However, in the yeast sequence the glycine is replaced by an alanine residue, SGNH**A**Q, and Tyr 119 (= Tyr121in hSR) is not observed in the core of the small domain. A glycine is conserved at this position in eukaryotes with a central nervous system, but not in yeast and plant SRs[27]. In the SR sequence alignment (Supplementary Fig. 7) residues that are within 3.8 Å of the side chain of Tyr 121 (S84, G88, Q89, T92, I104, A117, I118) in the 'open' structure (6ZUJ), are highlighted in red. Interestingly, the *C. elegans* SR (Uniprot code: Q93968_CAEEL) has the conserved glycine in the sequence SGNH**G**Q, but the equivalent residue to Thr 92 is an alanine.

The 'open' yeast spSR (1WTC) structure with the ATP analogue AMPPCP shows it binds at the dimer interface[37], virtually identical to the position we observe in our 'closed' hSR structure with malonate and ATP (Supplementary Fig. 8). Differences between 'open' yeast (1V71 and 1WTC) and 'open' hSR structures (5x2l, 6slh, 6zuj, 7nbc, 7nbd, 7nbf, 7nbg and 7nbh) means the ATP binding pocket does not exist in the 'open' hSR structure (Supplementary Fig. 2g).

**ATP and malonate cooperatively increase the thermal stability of hSR.** Thermal stability of hSR was assessed under different experimental conditions ( ± ATP, malonate). The reference Tm of SR with no additives (0 ATP, 0 malonate) was 47.3 ± 0.3 °C. Thermal denaturation curves (Fig. 4a) showed a negative (destabilising) thermal shift with 1 mM ATP and no malonate (ΔTm = −1.3 °C) and a positive (stabilising) thermal shift with no ATP and 1 mM malonate (ΔTm = +1.2 °C). The greatest

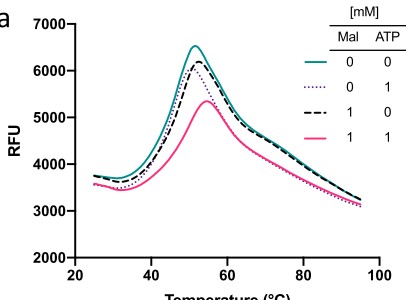
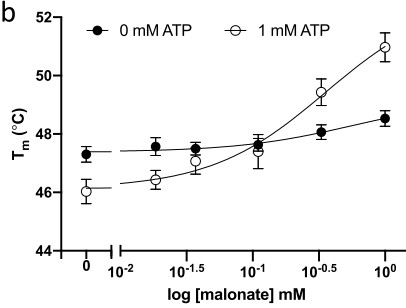

**Fig. 4 Thermal denaturation analysis of SR with ATP and malonate. a** Thermal denaturation curves of SR with and without 1 mM ATP and malonate. A right-shift indicates an increase in Tm and thermostability. The small right-shift and thermostability increase with the addition of malonate alone is augmented significantly when both ATP and malonate are present. **b** Melting temperatures of SR across a concentration series of malonate (0–1 mM) with and without 1 mM ATP. Tm shifts with malonate alone are negligible across the series; 1 mM ATP appears to have a destabilising effect at low concentrations of malonate before sufficient levels are reached to drive complex formation, resulting in a high Tm shift. The experiment was conducted in triplicate and repeated five individual times. Data points represent the mean of means, and error bars show standard error of the mean (Error bars = SEM, n = 5).

thermal shift generated was in the presence of both 1 mM ATP and 1 mM malonate ($\Delta$Tm = +3.7 °C).

Further analysis across a malonate concentration series (0–1 mM) showed that the basal thermal stability of SR with ATP at low concentrations of malonate (<0.1 mM) is reduced compared to that of SR without ATP (Fig. 4b). It is known that malonate exerts allosteric effects on the ATP site that increases the affinity of hSR for ATP by 100-fold and in turn, that ATP modulates malonate binding to the catalytic site and increases hSR affinity for malonate by at least 9-fold[54]. Moreover, it has been demonstrated that ATP binding to hSR induces a conformational change in the active site, decreasing its polarity and accessibility to solvent[45]. An H-bond network links the active and ATP sites helping to mediate the conformational transition from an open 'inactive' to a closed 'active' SR[37,45,55]. Our thermal stability data suggest that malonate concentrations >0.1 mM are required to achieve the cooperative stabilising effect of ATP and malonate. Thermal stability experiments on XChem compounds typically showed no significant shifts in Tm.

**Compounds 1-x0495 and 13-x0430 bind to different open conformations of the small domain.** The highly flexible small domain of hSR contains a small mobile subdomain (residues 78–81 and 101–148) linked to the large domain by four flexible regions (residues 68–77, 82–85, 99–101 and 149–151)[33]. The flexible region comprising residues 149–151 corresponds to the C-terminal β-strand of the small domain and for all three residues, changes in the phi-psi angles were observed between 'open' and 'closed' conformations of hSR[33]. In our previously published 'open' hSR structure (6SLH), two distinct positions were observed for the small domain in the C-subunit (Supplementary Fig. 1 in[33]), however in the optimised XChem crystal form, a single position is observed for the small domain of the C-subunit (see Methods for details).

In the hSR XChem structure bound to compound 1-x0495, the compound is observed in subunits A, B and D, but not in the subunit C (Fig. 5a, d). In contrast, the XChem structure with compound 13-x0430 showed the compound only bound to subunit C, with no compound observed in subunits A, B or D (Fig. 5b, c). The largest difference in phi-psi angles between A- and C-subunits of XChem crystals is around Met 150 (Supplementary Table 5), in the β6 strand (Fig. 5). In the A-subunit (Fig. 5a, b), the main chain NH of Val 151 donates a hydrogen bond to the main-chain C=O of Val 80 (from the β3-strand). However, in the subunit C the main-chain NH of Val 151 interacts indirectly with the C=O of Val 80 via a water (small red

sphere, Fig. 5c). This difference between A- and C-subunit conformations includes Tyr 121. In A-subunits, Tyr 121 (when not displaced by compound; Figs. 3, 5b) makes a direct (~3.2 Å) hydrogen bond with the main-chain carbonyl of Ser 84. However, in C-subunits the carbonyl of Ser 84 is too far from the Tyr 121 hydroxyl for this direct hydrogen bond to occur; instead, Tyr 121 is observed to interact indirectly with the main-chain carbonyl of Ser 83 via a water (small red sphere; Fig. 5c). The difference between the conformations of the A- and C-subunits can be seen when there is no compound bound to either subunit (Fig. 5d). When the central β-sheets of the small domains from A- and C-subunits are superposed, there are significant differences in the positions of the large domains, including Lys 56 and the covalently attached PLP (Fig. 5d). The differences in conformation between the 'open' A- and C-subunits in XChem crystals (Fig. 5d) are much smaller than those between the 'open' and 'closed' conformations (Supplementary Fig. 9, Supplementary Table 5).

**XChem compound binding sites at the dimer interface.** The dimer interface in hSR contains a highly conserved small β-sheet with two hydrogen bonds between the β-D strands from the two subunits (Fig. 2, see Supplementary Fig. 10 for views of hydrophobic core). There are two intersubunit hydrogen bonds, one from the carbonyl oxygen of Lys 279 to the NH of Leu 281' and one from the NH of Leu 281 to the carbonyl oxygen of Lys 279' (where' indicates the second subunit in the dimer; Fig. 2). Both hydrogen bonds are conserved between 'open' and 'closed' hSR structures and are also conserved in 'open' and 'closed' structures of more distant relatives such as yeast SR and human SDH.

This small β-sheet is conserved at the core of a partially flexible hydrophobic intersubunit contact (Supplementary Fig. 10), which allows some movement at the dimer interface (Supplementary Movie S1). In the XChem screen (Supplementary Tables 1 and 4), compounds 14-x0478 and 15-x0306 (pdb codes: 7NBD and 7NBF) were observed on either side of this β-sheet (Fig. 6).

The binding site for compound 14-x0478 is 'above' the intersubunit β-sheet but below C-terminal helical regions (Fig. 6a, b). Compound 14-x0478 binds in a hydrophobic pocket on the twofold axis 'above' Leu 281 (and 281') (view as in Fig. 6a) and between Trp 275, 275' (indicates the residue comes from the second subunit in the dimer). In most 'open' hSR structures, the electron density for the C-terminal region (between Leu 319 and Trp 325) is very weak, and residues 326–340 have little or no electron density so are not modelled. In the malonate/ATP 'closed' complex, the C-terminal residues are better ordered and

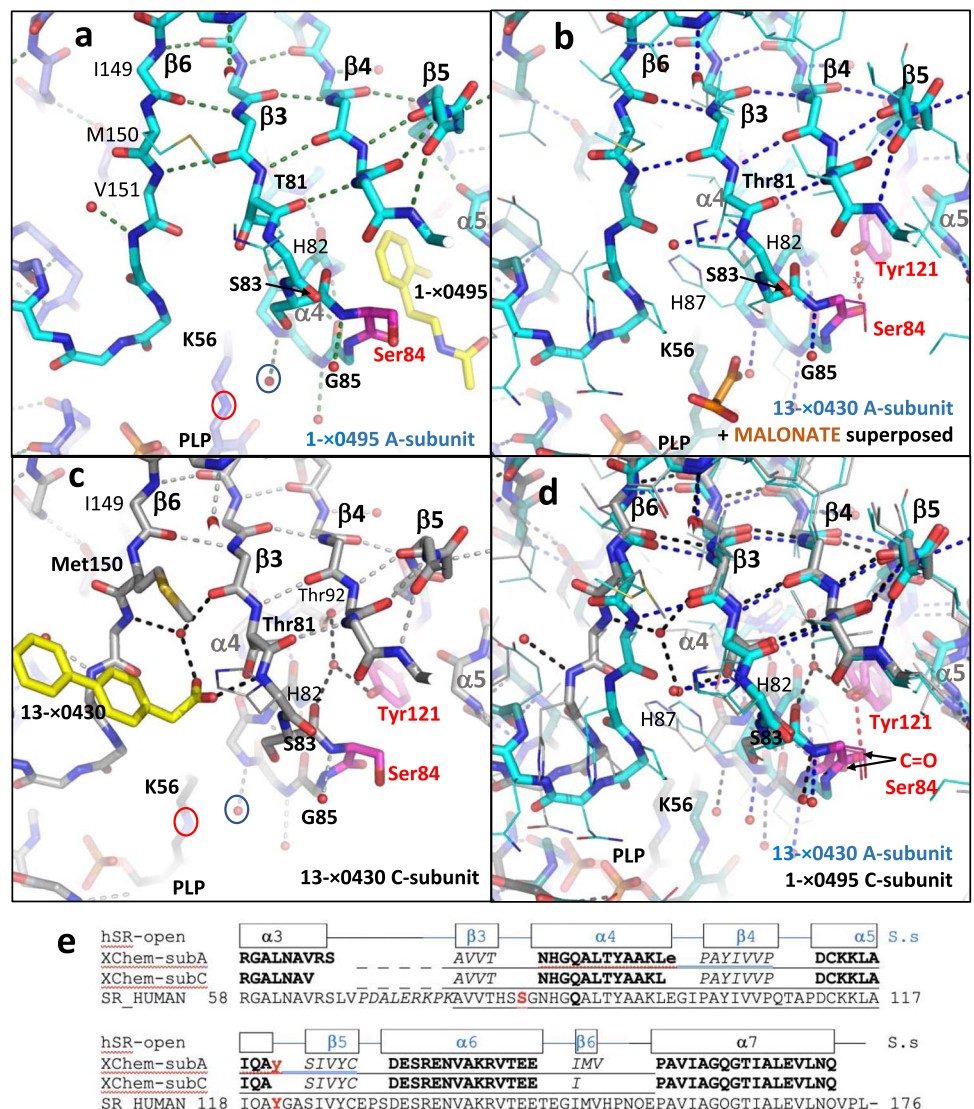

**Fig. 5 XChem compounds 1-x0495 and 13-x0430 bind to different open conformations of the small domain. a** View of the central β-sheet region of the A-subunit small domain in the 1.53 Å crystal structure of hSR with **1**-x0495 (7NBG; yellow carbons). All main-chain atoms are shown in stick (cyan/blue carbons) with selected side-chains shown in line (H82, M150) or stick (T81, S83). The catalytic residue Ser 84 is highlighted (magenta carbons). A water (small red sphere) and the nitrogen linking Lys 56 to the PLP are ringed. Hydrogen bonds (<3.3 Å) are shown as dotted lines. **b** Equivalent view of subunit A from the 1.71 Å structure with compound **13**-x0430 (7NBC) with all side-chains shown as lines. A 3.2 Å hydrogen bond (dotted red line) links the side-chain hydroxyl of Tyr 121 (magenta carbons) to the main-chain carbonyl oxygen of Ser 84. Compound **13**-x0430 is not observed in subunit A; malonate is superposed on 7NBC from the α4 helix of the ATP-malonate structure (6ZSP). **c** Equivalent view of subunit C from the 1.71 Å **13**-x0430 structure (7NBC). The carboxylate group of the compound accepts hydrogen bonds from a water, the side-chain hydroxyl of Thr 81, and the main-chain NH of His 82. Note Tyr 121 (magenta carbons) forms a hydrogen bond with a water that donates a hydrogen bond to the main-chain carbonyl oxygen of Ser 83. **d** Equivalent view of subunit C from the 1.53 Å **1**-x0495 structure (7NBG; grey carbons) and subunit A from the 1.71 Å **13**-x0430 structure (7NBC; cyan carbons) with all side-chains shown as lines. No compound is observed in either subunit. Note different positions of Ser 84 carbonyl oxygens (arrowed). **e** Sequence alignment showing different lengths of the β6-strand (149–151 or 149) in A and C subunits of XChem soaked crystals.

Trp 325 has good density and reasonable temperature factors. Residues after Val 326 diverge in the A and B-subunits with the last residues being Glu 330 in the A-subunit and Gln 328 in the B-subunit. The C-terminal four amino acids of hSR (337 SVSV 340) interact with both GRIP[56] and PICK1[25], and coexpression of GRIP with SR can cause an increase D-serine levels[26].

XChem compound **15**-x0306 binds 'below' the inter-subunit β-sheet (Fig. 6c). Two pockets are found on either side of the twofold axis, quite close to Leu 25 (or Leu 25′). Small movements at the dimer interface mean the pockets into which fragments x0478 and x0306 have bound in the 'open' structure do not exist in the 'closed' ATP/malonate structure (Fig. 6b, d).

**Compound 2-x0406 has two adjacent binding sites**. XChem compound **2**-x0406 binds in the Tyr 121 pocket in subunits A, B and D, and in all three subunits also has a second adjacent binding site (Fig. 7a, b). The compound is not observed in the subunit C. In subunits A, B, and D, one of the nitrogens from the benzimidazole donates a hydrogen bond to either: (i) the main-chain C=O of Ser 84, if occupying the Tyr 121 pocket, or (ii) to the side-chain of Asp 238, if occupying the adjacent binding site. The proximity of this second **2**-x0406 binding site to the Tyr 121 pocket suggests that any compound binding in the Tyr 121 pocket (Supplementary Table 3) could be linked to **2**-x0406 to increase the size and potency of the compound. The second **2**-

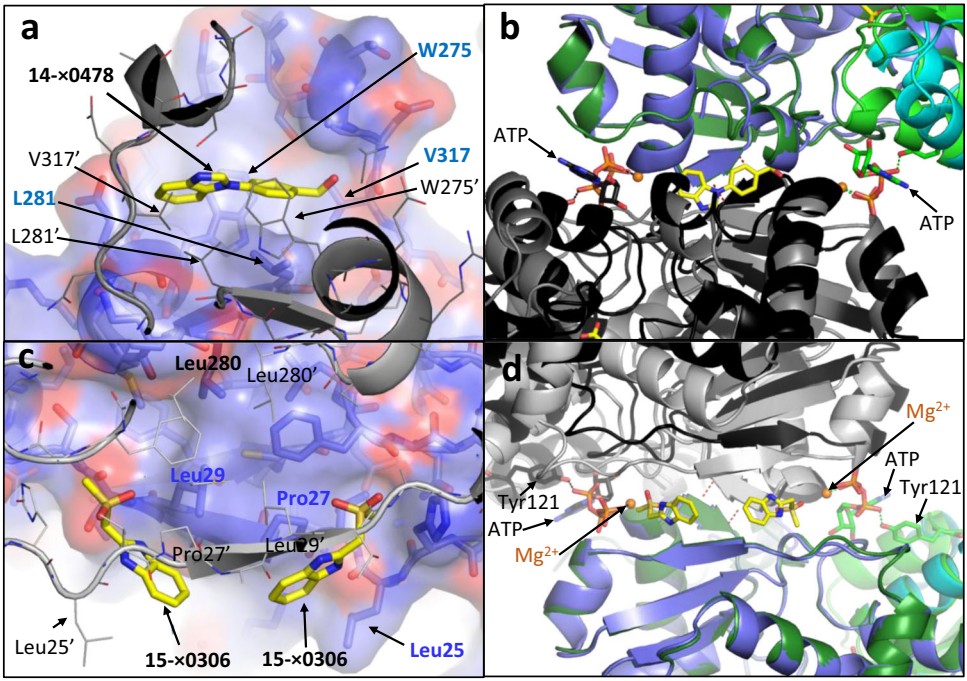

**Fig. 6 XChem compounds 14-x0478 and 15-x0306 bind at the dimer interface. a** View of 1.86 Å hSR structure with compound **14**-x0478 (7NBD; yellow carbons) binding on the twofold axis near the C-terminal helical regions of the A and C subunits. The A subunit (blue/cyan carbons) is shown with a semi-transparent surface and side-chains in stick representation; the side-chain of Trp 275 from the A subunit is behind the compound. The C-subunit in the foreground (grey cartoon) is shown with side-chains in line representation. **b** Orthogonal view of (**a**) with the closed malonate/ATP structure (A subunit in green, B subunit in black) superposed on large domain of the **14**-x0478 structure (A subunit in blue, C subunit in grey). Compound **14**-x0478 and malonate are shown in stick (yellow carbons). **c** View of the 1.60 Å hSR structure with compound **15**-x0306 (7NBF; yellow carbons) binding at two positions on either side of the twofold axis, close to Leu 25. Note Leu 280 and 280′ are at the core of the dimer interface, but are not close to compounds. **d** Orthogonal view from below (**c**) of the 'open' **15**-x0306 structure with the 'closed' malonate/ATP structure (A subunit in green, B subunit in black) superposed on the large domain of the A subunit (blue cartoon) of 7NBF. Note that in the 'open' **14**-x0478 structure, the β-strand (light grey) above the right-hand compound is much closer to the compound than the equivalent β-strand (black) in the ATP 'closed' structure.

x0406 binding site is at a distinct location from all other compound binding sites described in this paper (Fig. 7c). Interestingly, it is possible to dock one of the slow-binding peptidomimetic inhibitors identified by Dixon et al.[46] into both the Tyr 121 pocket and the adjacent **2**-x0406 binding site (Fig. 7d, e).

## Discussion

While there is a clear rational for developing SR inhibitors, there has been limited success. Perhaps the most promising compounds to date, which are derived from the slow-binding hSR inhibitors identified by Dixon et al.[46], are the novel serine racemase inhibitors reported to suppress neuronal over-activation in vivo[39,57]. It would be interesting to determine whether such compounds utilize the Tyr 121 pocket. In 2020, it was reported that the natural product madecassoside is an inhibitor of SR[49], although a supporting co-crystal structure has not yet been solved. While malonate and related acids (Supplementary Table 6) remain useful tool compounds, they are likely too polar to be developed as drugs.

The data in this paper are consistent with a model in which hSR is 'switched off' in the 'open' conformation. Closure of the small domain around the substrate to form the 'closed' conformation is favoured by binding of ATP (supplementary Fig. 11) at the dimer interface. However, Tyr 121 needs to 'flip out' in order for the small domain to close (Supplementary Movie S1). The observation that Tyr 121 is labile may be of some assistance in designing more potent and less polar inhibitors. Of note, the slow-binding SR inhibitory compounds identified by Dixon

et al.[46], which demonstrated greater inhibition after 15 min, all contained a terminal benzene moiety that is reminiscent of many of the twelve XChem compounds binding in the Tyr 121 pocket (Supplementary Table 3, 4; Table 1 and supplementary Fig. 12). While all twelve of these XChem compounds are able to bind and trap an 'open' conformation of hSR with Tyr 121 in the 'out' position (Fig. 3; Supplementary Table 3), one of them compound **2**-x0406 has a second binding site adjacent to the Tyr 121 pocket (Fig. 7).

The XChem screen described in this paper was performed using crystals in which SR was in an 'open' conformation. Interestingly, of the four subunits in the asymmetric unit, subunit C is locked in a slightly different open conformation (Fig. 5). and compound **13**-x0430, was only observed to bind to the C subunit (Fig. 5).

We also report two different binding sites for XChem compounds at the dimer interface (Fig. 6, compounds **14**-x0478 and **15**-x0306). Some movement at the dimer interface is observed when 'open' and 'closed' conformations are compared (Fig. 2, Fig. 6, Supplementary Fig. 10). Residues at the C-terminus of SR which are implicated in regulating activity may alter subunit-subunit interactions (see Supplementary discussion). The roles of Ser 84 and Lys 56 as the Si and Re-face acid/bases (Supplementary Fig. 13) in the catalytic cycle of hSR are generally accepted. However, despite the elucidation of many mechanistically informative crystal structures (Fig. 8, Supplementary Table 2), alternative residues have been proposed as hydrogen bond donors for some steps in the catalytic cycles of SR (Fig. 9, Fig. 10, Supplementary Fig. 14). The evolution and regulation of the four

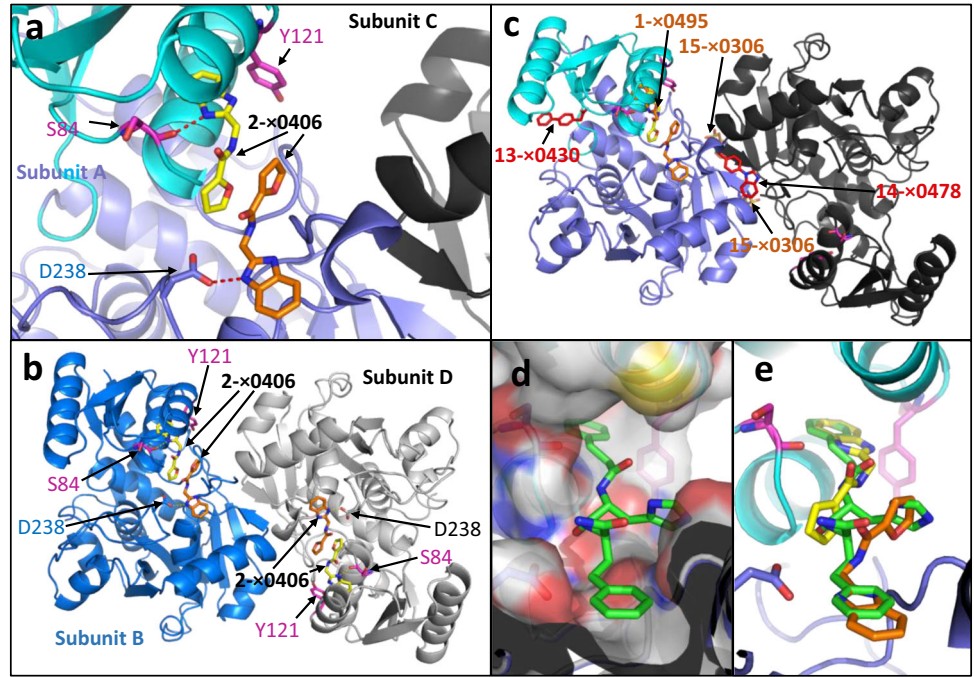

**Fig. 7 XChem compound 2-x0406 has two adjacent binding sites. a** The A subunit from the AC dimer in the 1.77 Å crystal structure of hSR with 2-x0406 (pdb code: 7NBH) contains two compounds. The compounds donate hydrogen bonds (dotted red-lines) to the main chain carboxyl of Ser 84 or to the side-chain of Asp 238. **b** The BD dimer with similar orientation as (**a**). Two molecules of compound 2-x0406 are also bound to the B and D subunits.
**c** Superposition on the AC dimer of 2-x0406 (7NBH) of four other XChem compounds (sticks, labelled) from 7NBD, 7NBF, 7NBG, and 7NBC (C-subunit superposed on A). Note the C-subunit has no 2-x0406 or 1-x0495 bound. **d, e** Compound **9** (green carbons) from Dixon et al. (2006)[46] has been docked into the electron density corresponding to two adjacent molecules of compound 2-x0406 (yellow and orange carbons).

| Table 1 Primer sequences used for mutants. | |
| --- | --- |
| **Primer name** | **Primer Sequence (length bp)** |
| Y121F Rev | CAATTGACGCTCCGAAGGCTTGTATTGCAAGTTTTTTACAGT (42) |
| Y121F For | ACTGTAAAAAACTTGCAATACAAGCCTTCGGAGCGTCAATTG (42) |
| Y121A Rev | CACAGTATACAATTGACGCTCCGGCGGCTTGTATTGCAAGTTTTTTAC (48) |
| Y121A For | GTAAAAAACTTGCAATACAAGCCGCCGGAGCGTCAATTGTATACTGTG (48) |
| Y121G Rev | CACAGTATACAATTGACGCTCCGCCGGCTTGTATTGCAAGTTTTTTAC (48) |
| Y121G For | GTAAAAAACTTGCAATACAAGCCGGCGGAGCGTCAATTGTATACTGTG (48) |

reactions, on L- or D-serine, catalysed by hSR are discussed in more detail in the Supplementary discussion. One possibility could be that the two different conformations of the small domain we observe in the A and C subunits in 'open' XChem crystal structures, favour binding of different substrates (L or D-serine).

It is interesting that, although SR functions as a dimer, all the active site residues are located in a single subunit. Serine racemase may have evolved from a serine dehydratase enzyme[19] (see supplementary discussion and supplementary Fig. 15). It is possible that, for one or two of the four reactions catalysed by SR, the purpose of the second subunit in the dimer may be to help guide the 'active' subunit in the dimer along a particular catalytic pathway. We have made three movies illustrating structural changes between 'open' and 'closed' conformations (Supplementary Movies S1, S2, S3).

An understanding of how the different catalytic activities of hSR are regulated in the brain may prove useful in guiding drug discovery against this fascinating, but as yet, undrugged enzyme.

## Methods
All materials were obtained from Sigma unless otherwise stated.

**Serine racemase wild-type and Y121 mutations, expression and purification**. Human SR containing a C-terminal polyhistidine (His)$_6$ tag and two cysteine-to-aspartate point mutations (C2D, C6D) to improve enzyme stability and solubility during the purification process[31], was expressed and purified as described[33].

Tyrosine mutations (Y121A, Y121G, Y121F) were introduced in the pET24a_SR vector by site directed mutagenesis using *PfuUltra* high-fidelity (HF) DNA polymerase and *Dpn*I according to the QuikChange protocol (Agilent). The mutagenic oligonucleotide primer pairs (see Table 1) were designed using the QuikChange Primer Design (https://www.agilent.com/store/primerDesignProgram.jsp) and these were synthesised and HPLC purified by Euofins Genomics.

The *Dpn*I-treated DNA was transformed into One Shot™ TOP10 Chemically Competent *E. coli* (ThermoFisher) and plated on LB agar with kanamycin at 50 μg/ml. Three colonies for each mutation were selected and vector was purified using the ISOLATE II Plasmid Mini Kit (BIOLINE). The single amino acid mutations were confirmed by sequencing (Eurofins Genomics). One vector of each mutation was then transformed into the expression strain BL21-CodonPlus (DE3)-RIL competent cells and expressed and purified as described[33].

For crystallisation and thermal denaturation experiments, hSR-containing fractions were concentrated to 15 mg/mL (0.40 mM) in 20 mM Tris-HCl pH 8.0, 100 mM NaCl, 10% glycerol (Fisher), 1 mM MgCl$_2$, 0.5 mM ATP, 50 μM PLP and 5 mM DTT (Fisher) before being flash frozen in liquid nitrogen and stored at −80 °C. The protein concentration was determined by UV spectrophotometry at 280 nm using a molar extinction coefficient of 29910 and a molecular weight of 37.4 kDa. The protein yield was approximately 3 mg/L and was over 95% pure.

Protein for assays, hSR and the three Y121 mutated proteins hSR Y121F, Y121A and Y121G were concentrated in 20 mM Tris-HCl pH 8.0, 100 mM NaCl, 10% glycerol, 1 mM MgCl2, 0.5 mM ATP, 50 μM PLP and 5 mM DTT. The

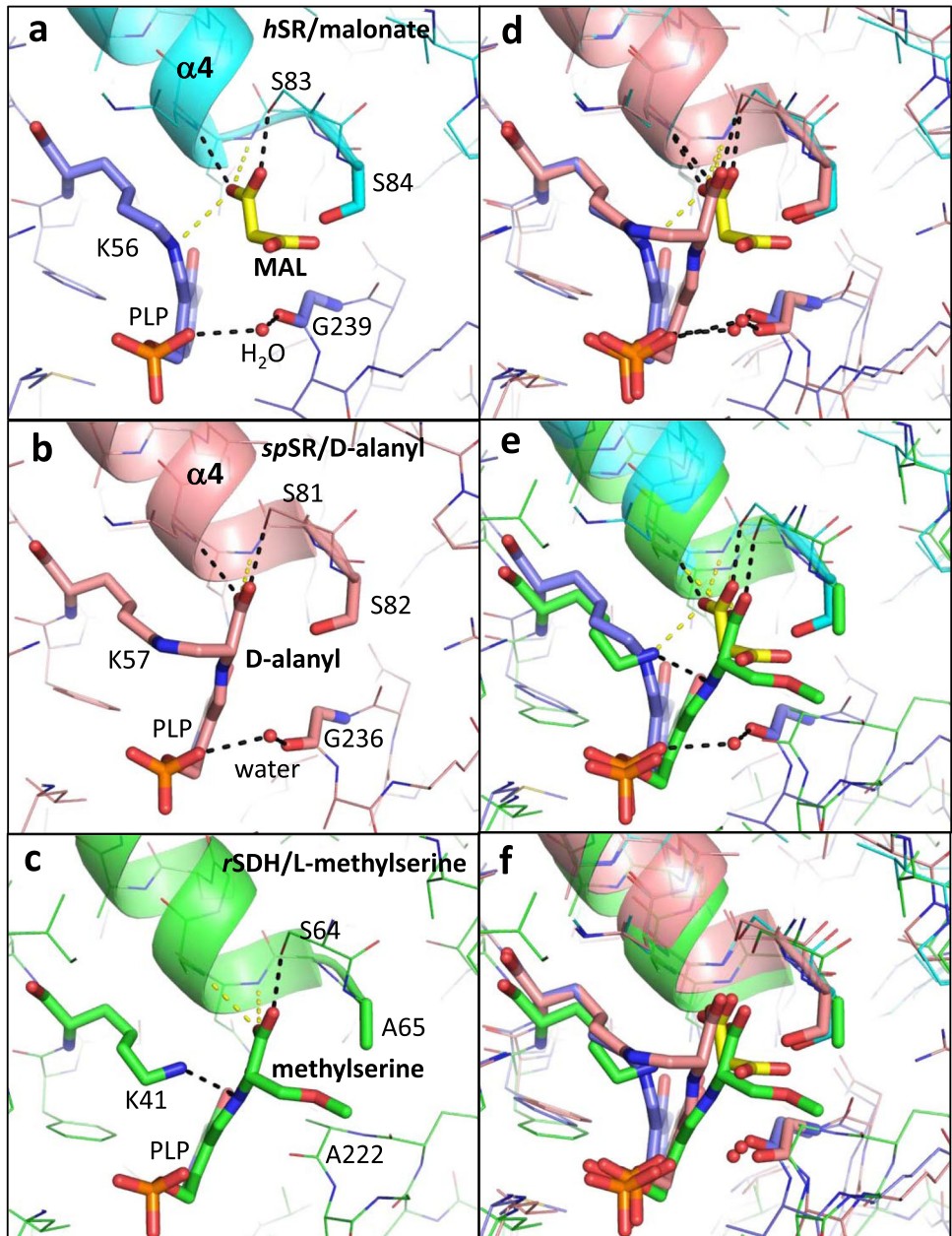

**Fig. 8 Structural comparison of human SR with malonate (6ZSP), d-alanyl yeast SR (2ZPU) and rat SDH with methylserine (1PWH). a** 1.60 Å structure of hSR with malonate (6ZSP). H-bonds are indicated by dotted black lines (2.5–3.0 Å) or dotted yellow lines (3.01–3.29 Å). A water molecule (small red sphere) makes H-bonds to the main-chain carbonyl of Gly 239 and the phosphate from PLP. **b** 1.70 Å structure of *S. pombe* SR with lysino-D-alanyl covalently attached to the PLP (2ZPU). **c** 2.60 Å structure of rat L-serine dehydratase complexed with O-methyl-serine covalently attached to PLP (1PWH). **d** Comparison of (**a**) and (**b**). Note the conserved water making an H-bond to the phosphate and the glycine residue. **e** Comparison of (**a**) and (**c**). Note Ala 65 is equivalent to Ser 84. **f** Comparison of (**a**), (**b**) and (**c**).

protein concentration was determined using Quick Start™ Bradford Protein Assay (Biorad). The mutated proteins were confirmed by Liquid chromatography-mass spectrometry (LC-MS) on a Waters Synapt G2- Si quadrupole time-of-flight mass spectrometer coupled to a Waters Acquity H-Class ultraperformance liquid chromatography (UPLC) system at Cardiff University. Spectra were collected in positive ionisation mode and analysed using Waters MassLynx software version 4.1. Deconvolution of protein charged states was obtained using the maximum entropy 1 processing software. Experimentally determined masses for the three mutants, Y121F (37397), Y121A (37321) and Y121G (37306.5), were as expected.

**Crystallisation and crystal soaking experiments.** Human malonate-bound SR was crystallised at 20 °C in 48-well MRC Maxi sitting-drop plates (Molecular Dimensions) in 25% PEG 3350, 230 mM sodium malonate, and 15 mM magnesium

chloride using 5 mg/mL purified SR plus fresh 5 mM DTT equilibrated against 100 µL reservoir. The drops were dispensed in a 1:1 ratio (500 nL protein + 500 nL reservoir). Single crystals were harvested and cryo-protected sequentially in 10, 20, and 30% glycerol before storing in liquid nitrogen. Additional ATP was not added in the crystallisation experiment, but ATP in the protein delivery buffer was present at 1.25-fold times the protein (subunit) concentration and MgCl$_2$ was present at 2.5-fold the protein (subunit) concentration.

A crystallographic fragment screen was performed on *holo* crystals using the XChem screening strategy at Diamond Light Source[58]. For the XChem experiments, *holo* crystals were generated at 20 °C in 96-well Swissci 3-drop, sitting drop plates (Molecular Dimensions) using 14 mg/mL SR plus fresh 5 mM DTT dispensed in a 3:2 ratio (300 nL protein + 200 nL reservoir) equilibrated against 20 µL reservoir containing 100 mM MES pH 6.2, 100 mM Calcium chloride, 5% ethylene glycol and 20% PEG Smear Broad (Molecular Dimensions). This crystallisation condition differs from that reported in a recent crystal structure of

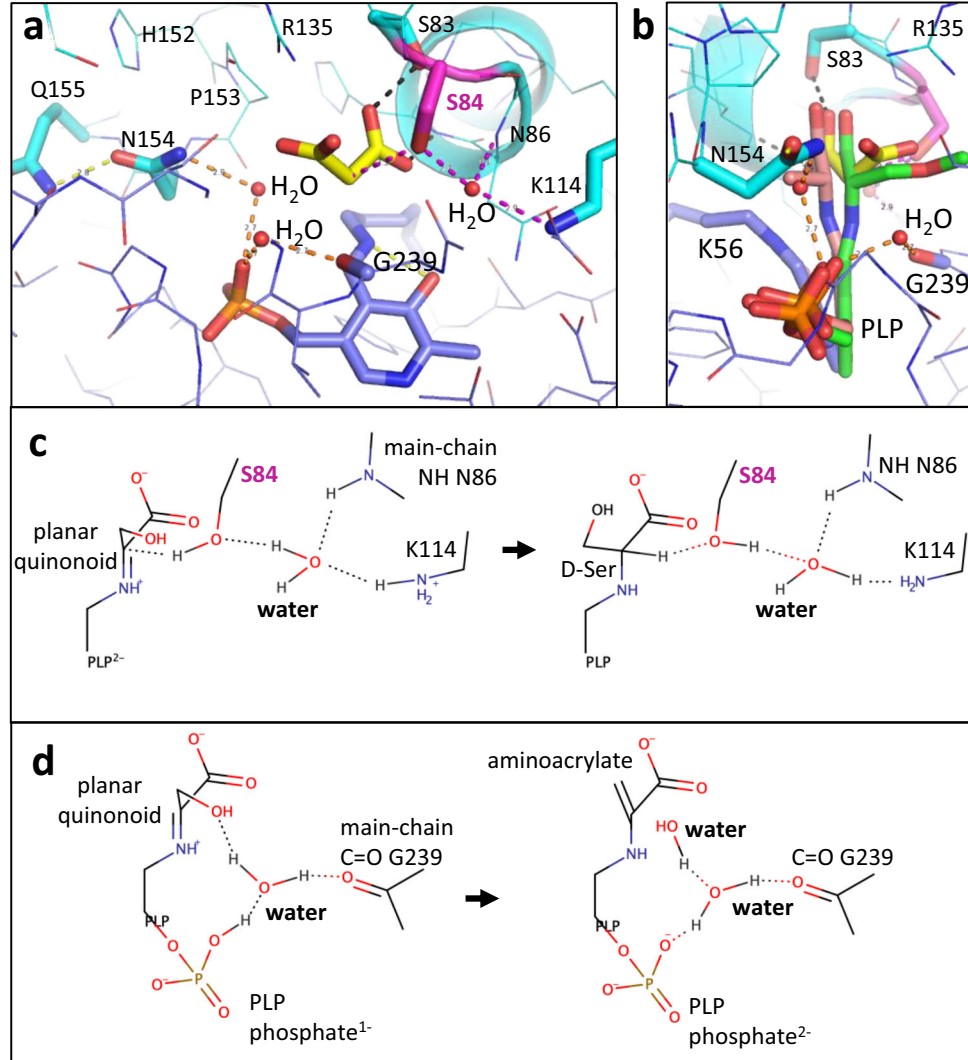

**Fig. 9 Comparison of proposed mechanisms for racemisation and β-elimination. a** View of the active site in the malonate (yellow carbons) hSR complex (6ZSP). Three water molecules are shown (small red spheres). **b** Orthogonal view with D-alanyl-PLP (pink carbons) and methylserine PLP (green carbons) superposed as in Fig. 8. **c** A water molecule between Lys 114 and Ser 84 has been proposed to facilitate protonation of the substrate to give D-serine[36]. **d** The dehydration of the substrate has been proposed to be facilitated by the phosphate group of PLP[79]. The mechanism shown is modified to include a water molecule.

*holo* SR (6SLH) which was crystallised against reservoir/precipitate of 100 mM Bis-Tris pH 6.5, 15% PEG 3350, 250 mM MgCl₂[33]. ATP was present in the protein delivery buffer at 1.25-fold times the protein subunit concentration.

Crystal soaking, harvesting, mounting, and data collections were performed at the Diamond Light Source XChem facilities and beamline I04-1 through the XChem workflow[50,59]. During preliminary crystal characterisation *holo* crystals were observed to tolerate concentrations of up to 40% DMSO. The full compound screen was performed on 600 crystals which were each soaked with 150 mM fragment compound (30% v/v of 500 mM stocks in DMSO) for ≥1.5 h before crystals were harvested and mounted on the fast microscope crystal mounting system[60]. The fragments screened were obtained from the Diamond-SGC-iNext Poised (DSiP) XChem library.

**Data collection and processing.** For the malonate/ATP-bound structure, an X-ray dataset was collected on a single cryo-cooled crystal at Diamond Light Source synchrotron (Supplementary Table 1) using the rotation method and a Pilatus detector at a wavelength of 0.978 Å (beamline I03). From the XChem pipeline, a *holo* crystal soaked in 20% DMSO gave a 1.80 Å dataset using the X-ray centring method at a wavelength of 0.916 Å (beamline I04-1). The diffraction images were processed using the Xia2 software pipeline[61]: DIALS[62] for data processing, POINTLESS and AIMLESS[63] for symmetry determination and scaling, and CTRUNCATE[64] for calculation of amplitudes and intensities. Data from other XChem crystals (Supplementary Table 1) were collected and processed as for the XChem *holo* crystal.

**Structure solution and refinement.** Processing and refinement were carried out using the CCP4[65] and phenix[66] suite of programmes. The structure of the hSR complex with malonate and ATP (pdb code: 6ZSP) was solved by molecular replacement in MOLREP[67] using the crystal structure from the previously determined malonate-bound structure (pdb code: 3L6B) with iterative refinement in REFMAC5[68] and manual model building in Coot[69]. ATP, malonate, waters, and other ligands were initially identified in the difference (Fo-Fc) electron density maps and refined. The final 2Fo-Fc maps showed clear density for ATP and malonate which was refined with full occupancy and have B-values similar to surrounding residues[70].

The structure of the representative XChem *holo* crystal (20% DMSO soak, no fragment) presented in this paper (6ZUJ) was solved by rigid body refinement based on a previous structure (6SLH)[33] and refined with phenix.refine[71] and REFMAC5[68,72] and modelled in coot[69]. Initial analysis of approx. 600 ligand-soaked crystals was carried out using XChemExplorer[59] a data-management and workflow tool to support large-scale simultaneous analysis of protein–ligand complexes during structure-based ligand discovery. The workflow flagged 95 structures with binding events, 44 of which were classified as interesting, spanning four distinct binding sites. Five SR structures from the XChem screen are described in this paper that are representative examples of the four major binding sites. One compound, **2**-x0406, occupied not just one of the four major binding sites, but also occupied a second site adjacent to the tyrosine 121 pocket.

The five XChem fragment-bound crystals presented here were refined from the DMSO control structure (6ZUJ, Supplementary Table 1) and/or other XChem structures, using REFMAC5[68,72] and coot[69,73], ligand geometry and restraints were

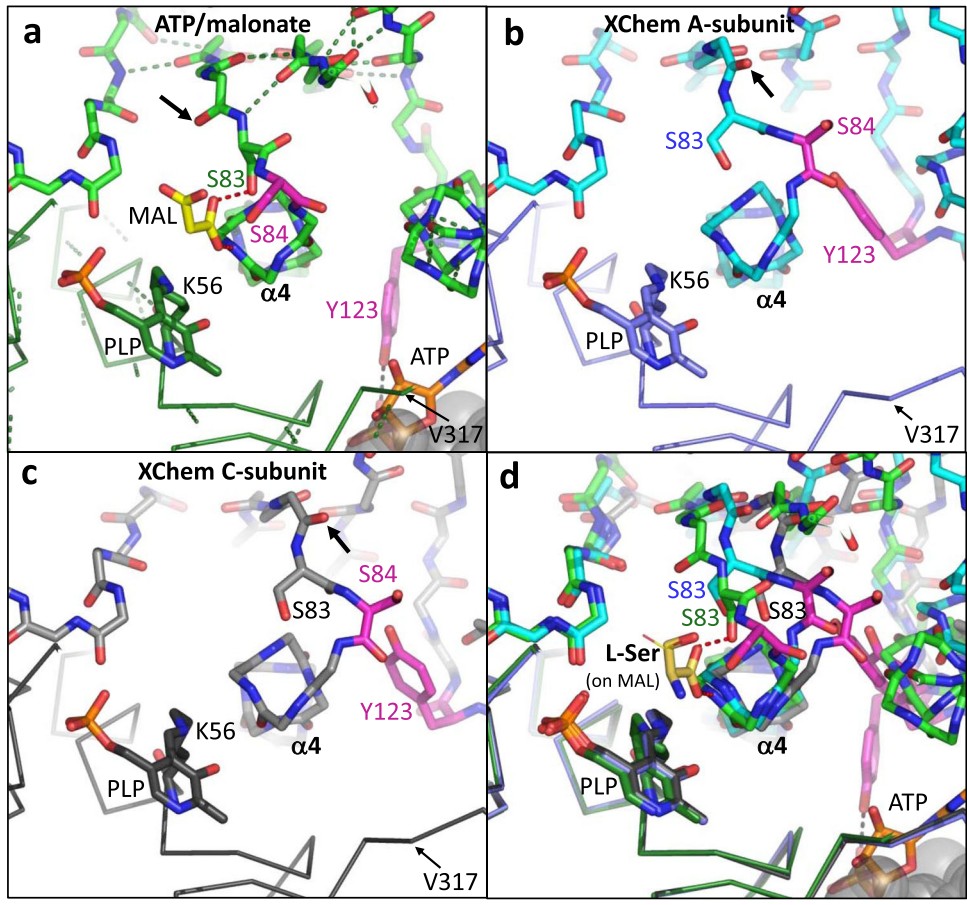

**Fig. 10 Comparison of human SR structures viewed from N-terminal end of α4 helix. a** View of the malonate (yellow carbons)/ATP (oranges carbons) hSR complex (6ZSP). SR (green carbons) is shown with side-chains of Ser 84 and Tyr 121 (magenta carbons) highlighted. The second subunit in the dimer is partially visible in the bottom right-hand corner (grey semi-transparent spheres). The side-chain of Ser 83 makes a hydrogen bond (dotted red line) to the malonate and the main-chain NH of His 87 also donates a hydrogen bond to the malonate. The carbonyl oxygen of His 82 is arrowed. **b** Equivalent view of an XChem A subunit. The carbonyl oxygen of His 82 is arrowed. **c** Equivalent view of an XChem C subunit. **d** Superposition of structures from (**a**), (**b**) and (**c**) on the large subunit (Cα traces). L-serine (sticks - yellow carbons) has been modelled on the malonate (thin yellow lines - largely hidden under L-serine).

generated with AceDRG[74]. Ligand occupancies for XChem compounds were initially estimated from electron density maps (Fo-Fc Y121A omit maps for Tyr 121-displacing structures) and refined B-factors were used to check final ligand occupancies were reasonable[70]. All structures have good geometry (Supplementary Table 1) and clear electron density for ligands (Supplementary Figs. 16 and 17).

**Notes on metal identification and chemical nomenclature**. XChem crystals were grown in the presence of 1 mM MgCl2 and 100 mM NaCl (protein delivery buffer) and 100 mM calcium chloride (crystallisation buffer). Sodium (Na⁺), and magnesium (Mg²⁺) ions have ten electrons, the same number as waters, and were identified largely by their coordination geometry[75]. Calcium ions (Ca²⁺) have 20 electrons and have larger peaks in electron density maps. The refined structures from the XChem screens each contains four calcium ions, one in each subunit at the metal binding pocket near to the phosphate of the PLP. In all structures the PLP is covalently attached to lysine 56 (see Supplementary Fig. 1). We note that although, in kekule representation, a Cα carbanion is sometimes drawn - QM calculations suggest that 'the electrons on Cα formal carbanion are delocalized throughout the PLP cofactor'[76]. We have not used the formal Cα carbanion description in this paper.

The XChem compounds discussed in this paper together with inhibition data are shown in Supplementary Table 4. Chemical structures were drawn with MarvinSketch 20.20 (https://www.chemaxon.com), which was also used to predict tautomers and charged states of compounds.

**Activity assays**. The ability of compounds to inhibit the L to D-Ser racemisation activity of hSR was measured in a coupled assay. In the assay, hSR converts L-Ser to D-Ser. The D-Ser is then converted into pyruvate, ammonia and hydrogen peroxide using D-amino acid oxidase. Next, the hydrogen peroxide reacts with luminol in the presence of horseradish peroxidase, producing luminescent byproduct. In the assay malonate was used as a control compound and gave 100%

inhibition – results for XChem compounds are shown in Supplementary Table 4 and in supplementary Fig. 12. The activity of the wild-type and the three Y121 mutants, Y121F, Y121A and Y121G was measured at six ATP concentrations (1 mM, 0.5 mM, 0.25 mM, 0.125 mM, 0.062 mM and 0.031 mM) as shown in supplementary Fig. 11.

The inhibition of hSR activity was measured using a luminescence assay in a 384 well plate (flat bottom white, Corning). Initially, each well contained 100 mM Tris pH 8.5, 0.05% Pluronic f-127 (VWR), 1 mM MgCl2, 1 mM ATP (disodium salt, Fisher), 10 µM PLP, 0.3 nM hSR, 2 mM L-serine and compound at 16.67 mM to 0.0077 mM (see supplementary Fig. 12) and had 16.7% DMSO concentration. The plate was incubated at RT with shaking for 30 min. After this, the reaction was stopped using a detection mix. The final well concentration was 100 mM Tris pH 8.5, 0.05% Pluronic f-127 (VWR), 1 mM EDTA (Fisher), 0.25 U/mL horseradish peroxidase, 0.5 U/mL D-amino acid oxidase and 25 µM Luminol (Insight Biotechnology) and 8.4% DMSO. The plate was left to incubate covered and shaking for 10 min. The plate was then read using a PHERAstar (BMG) plate reader. For the ATP dependant activity data, the method was kept the same, except for varying the concentration of ATP.

The relative light units read (x) on the PHERAstar were converted into percentage inhibition (PI) using the equation:

$$PI = 100x[1 - (x - Min)/(Max - Min)]$$

where min is negative control (100% inhibition) and max is the positive control (no inhibition). The IC50s were calculated with GraphPad (Prism 8.3.1, Graphpad Software, San Diego CA). For compounds where inhibition was seen at 16.7 mM and 5.55 mM, but this was less than 100% inhibition (see supplementary Fig. 12), an extra 'artificial' data point with 100% inhibition at 1 M compound concentration was introduced for calculation of IC50s by GraphPad - such compounds have a ~ in front of the values in supplementary table 4.

**Thermal denaturation assay.** The melting temperature ($T_m$) of SR was determined in the presence and absence of malonate (disodium salt) and 1 mM ATP (disodium salt, Fisher) in a buffer containing 100 mM CHES pH 8.5, 1 mM $MgCl_2$, and 10 µM PLP. The reaction mixture was prepared in 96-well plates containing final concentrations of 2 µM SR, 5X SYPRO Orange dye (from 5000X concentrate, ThermoFisher), and a 6-point 1:3 dilution series of malonate (0–1 mM). Each condition was assessed with and without 1 mM ATP in the buffer. Thermal denaturation analysis was performed in triplicate by transferring 10 µL reaction mixture to 384-well plates for use in a BioRad CFX384 Touch Real-Time PCR Detection System. Programme: 10 min hold at 25 °C, plate read, increase by increments of 0.5 °C up to 95 °C; each step accompanied by a 15 s hold and plate read. Data were analysed using GraphPad Prism.

**Structural analysis.** Secondary structures were initially calculated with the PDBSUM server[77] before manual adjustment based on analysis of hydrogen bonding patterns. Of note, we define a small intersubunit β-sheet consisting of two anti-parallel β-strands, one from each subunit, which we named β-D (D for dimer). This sheet contains two hydrogen bonds, one from the carbonyl oxygen of Lys 279 to the NH of Leu 281′ and one from the NH of Leu 281 to the carbonyl oxygen of Lys 279′ (where′ indicates the second subunit in the dimer; see Fig. 2, Supplementary Fig. 10). This intersubunit β-sheet is conserved between 'open' and 'closed' structures and is at the core of a partially flexible hydrophobic intersubunit contact (Supplementary Fig 10). It also appears to be evolutionarily conserved and is present, for example, in SDH crystal structures (PDB codes: 1PWE and 1PWH; in 1PWE there are three dimers in the asymmetric unit and subunits need to be symmetry-flipped to see this 'conserved' dimer interface).

The small domain of hSR was modelled in two positions in the C subunit of a previous (pdb code 6SLH) structure (Supplementary Fig. 1 from Koulouris et al. 2020[33]). The temperature factors of the small domains (78–151) are relatively high in 6SLH (57.1, 100.4, 76.9 and 66.4 Å²). The crystals grown for the XChem experiments (6ZUJ) are in the same $P2_1$ cell as 6SLH[33] and have the same four subunits in the asymmetric unit, but the small domain of the C-subunit is well ordered and in a single well-defined conformation. The temperature factors of 6ZUJ (33.4, 66.9, 49.5 and 49.0 Å² for residues 78–151 of chains A, B, C and D) are lower than those for 6SLH. The small domain is composed of a four-stranded β-sheet and three α-helices. Not only is the first strand (78–81 or 78–83) in the small domain of variable length[33], but the adjacent last strand (β6) is shorter in subunits C and D, lacking the H-bond normally observed between the main-chain NH of Val 151 and the carbonyl oxygen of Val 80 (Fig. 5).

Figures depicting protein structures were drawn with PyMol (v.1.5.0.4; Schrodinger) which starts and ends β-strands at the Cα position of a residue. This approach is suitable for Val 151 at the C-terminal end of the β6-strand as the NH but not the C=O is involved in a β-strand H-bond, but it is not suitable for Ile 149 as both the NH and C=O of Ile 149 are involved in β-sheet hydrogen bonds. The previous residue, Gly 148, is not in the β-strand which should be drawn starting at the beginning of Ile 149 (Fig. 5).

**Modelling single conformations and making movies.** SR is an inherently flexible protein and electron density maps showed multiple conformations for several sidechains. Before making movies, structures were edited to remove multiple sidechain conformations, and where possible structures involved in movies were edited to start with analogous sidechain conformations. Otherwise, the sidechains selected for deletion were those that displayed the lowest occupancy (if different) or highest B-factor (if two occupancies of 0.5). Nomenclature inconsistencies were resolved by realigning chemically equivalent atoms based on conformation.

Three movies describing movement from the 'open' conformation (6ZUJ) to the 'closed' conformation (6ZSP) have been made; the first shows the overall dimer movement (Supplementary Movie S1), the second highlights movement of tyrosine 121 (Supplementary Movie S2) and the third shows structural changes in the small domain β-sheet that facilitate movement of Ser 84 (Supplementary Movie S3). To make the movies the large domain of the A subunit from the 'open' structure was superposed onto the large domain of the A subunit from the malonate/ATP structure. Then malonate was introduced into A subunit of the 'open' conformation structure by superposing the N-terminal end of the α4 helix (Asn 86 and His 87) onto open structures and 'copying' the malonate (see Fig. 10). Movies were created using PyMOL (The PyMOL Molecular Graphics System, Version 2.0 Schrödinger, LLC), and intermediate coordinates were generated using the ProSMART library for structure analysis[78]. Applying hierarchical aggregate transformations to residues, backbone and side chain atoms results in a parsimonious morphing from the open to the closed malonate conformation.

**Reporting summary.** Further information on research design is available in the Nature Research Reporting Summary linked to this article.

## Data availability

The coordinates and structure factors have been deposited in the Protein Data Bank. Details of the structures with accession codes (6ZSP for Mal/ATP) and (6ZUJ XChem apo, 7NBC, 7NBD, 7NBF, 7NBG, 7NBH, XChem fragment soaks) are listed in Supplementary Table 1.

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

## Acknowledgements

We are grateful to Diamond Light Source and all the staff that support the XChem laboratory and beamline I04-1, including Ailsa Powell, Romain Talon, and Alice Douangamath. The XChem fragment screen and downstream data analysis was supported by funding from a Medical Research Council Confidence-in-Concept grant. We thank the University of Sussex and the University of Cardiff for providing staff funding and laboratories for follow-up research.

## Author contributions

J.R.A. was the principal investigator and initiated the project. S.M.R. supervised and coordinated the research, analysed the XChem data, and refined several structures. C.R.K. and B.D.B. refined several structures, co-authored, reviewed, and edited all sections of the manuscript. C.R.K. contributed protein production, crystallisation, XChem sample preparation, pre-screen testing, and data analysis. T.K.H. performed thermal denaturation experiments and S.E.G.

preformed assays with supervision from O.G. K.T.E. made and purified Y121 mutant and wt proteins. J.G. and S.W. provided medicinal chemistry insight. O.G. supervised and assisted S.E.G. and T.K.H. and provided useful comments on the writing of the manuscript. R.A.N. produced chemically sensible movies. B.D.B. contributed refinement of all structures presented here. All authors read and approved the paper.

## Competing interests

The authors declare no competing interests.
