## [Peer Review File · Communications Biology]

Reviewers' comments:

Reviewer #1 (Remarks to the Author):

The manuscript by Koulouris et al. describes the investigation of a series of human serine racemase (hSR) structures obtained in a fragment screening campaign. One structure was determined for the first time as the "closed" crystal structure of hSR bound to ATP. In addition of a holo structure of hSR in the "open" conformation, five "open" structures bound to fragments were determined. A key role in the catalytic mechanism is assigned to Tyr121. Two fragments force the sidechain of Tyr121 into an out position and thus suggest a druggable pocket.

The manuscript is thoroughly written, giving a very detailed introduction, also including a comparison to corresponding yeast structures. I have no major objections, only a few minor comments:

1. Unless it is done during the editorial process, the manuscript should be carefully proofread to correct for several typos.
2. On page 17, line 340, the authors describe the position of compound 14 above Leu281. This residue should also be labelled in the figure.
3. Reference 32: The full reference including pages should be given.
4. Reference 48 and 55 are the same.
5. I do not like some of the figures; however, this is my personal opinion. For example in Fig. 3c, why is the semi-transparent surface shown? This distracts from the intended superposition of the molecules. Similar, what is the purpose of the multi-coloured semi-transparent surface of Fig. 6a and 6c? Here, the important part is the position of the bound fragment, which is hard to recognize in 6a and poorly visible in 6c. Instead a close-up view of the ligands with interacting residues should be shown. Same for Fig. 7a and 7b. The legend of Fig. 7b describes hydrogen bonds to Ser84 and Asp238, which I do not recognize in the figure.

Supplemental Material:

6. Table S1, second column: Instead of the 2D drawing, I would include the compound number, as the 2D description is also given in Table S2.
7. Table S1: Are the cell dimensions determined with different accuracies? If not, they should have the same number of digits after the period.
8. Table S4: How can the % inhibition be above 100% at 10 mM concentration? Also, is there an explanation why for compound 6, inhibition at 1mM is larger than at 10mM, assuming values above 100% are meaningful in the first place?
9. References: Check references 10, 23, 30, 50 and 54. I doubt these authors have only single character names. Reference 31 and 41 are the same.

Reviewer #2 (Remarks to the Author):

The manuscript by Koulouris et al. reports on the investigation of the PLP-dependent enzyme serine racemase exploiting both structural and kinetic methods. Results led authors to propose a key role for Tyr121 in the enzyme activation by coupling ATP site with the active site. In addition, authors have identified a novel class of enzyme inhibitors acting by displacing Tyr121 and stabilizing an open, inactive enzyme conformation.

Overall, the study has been carried out rigorously, nicely presented and discussed. A long waited structure of the human enzyme in the presence of the allosteric effector ATP has been determined. A few points should be addressed:

Line 141 . Quinoid should read quinonoid.

Line 354-356 Authors might rephrase the sentence that it is unclear with too many "closer", "closed".

Line 419-422 "It seems unlikely that both subunits in the dimer bind serine and catalyse reactions concurrently. The purpose of the second subunit in the dimer may be to help guide the 'active' subunit in the dimer along a particular catalytic pathway." This is indeed a strong statement as it implies a half of the site reactivity for serine racemase that has never

been reported. The same is true for all enzymes belonging to fold type 2 of the PLP-dependent family. Authors cannot derive only from structural data functional implications (see also below). Unfortunately, this line of thinking is still very popular among protein crystallographers. A structure does not tell us almost nothing on function. Efforts should be devoted to collect structural and functional data, potentially in the same physical state either in the crystalline state via microspectrophotometric studies or in solution via cryo-EM, and correlate them.

Line 525. The use of PMP, instead of PLP, seems quite odd and absolutely confusing. In literature and biochemistry textbook PLP is the active form of vitamin B6 that binds to enzyme active sites via a reversible Schiff base (or internal aldimine). PMP stands for the derivative that is formed in the transamination reaction upon transfer of the alpha amine of an amino acid to the internal aldimine. I suggest to retain the term PLP or, otherwise, reports as a footnote, IUPAC indications. If any, of the change in terminology.

Authors are proposing a key role for Tyr121 on the basis only of structural observations, i.e. the different conformation of Tyr121 in the absence and presence of ATP and potential inhibitors that displace Tyr121 from its position in the closed conformation. However, a structural observation that not necessarily translates in the activation/inhibition of the enzyme. To prove the point authors should substitute Tyr121 and determine the effect on ATP activation, as it is usually carried out in enzymology, and specifically on serine racemase (Canosa et al.) for assessing the role of Gln79. On this basis, even the paper title is partially misleading as the effective role of Tyr121 in the enzyme activation is not proved.

With the determination of human ATP-bound serine racemase structure authors are in the position to nicely discuss recent studies on the role of Cys113 nitrosylation (Marchesani et al, BBA, 2018; Marchesani et al. Febs Journal, 2020) as well as other previous studies of enzyme inhibition by NADH derivatives (Bruno et al., BBA, 2016) and ATP activation (Marchetti et al., 2013).

IC50s for the inhibitors for which structures are reported should be determined.

Authors did not try any structure-activity relationship on the list of identified inhibitors. This might be of help for the improvement in affinity.

Reviewer #3 (Remarks to the Author):

Human serine racemase(hSR)is the enzyme responsible forthe biosynthesis of the important NMDA receptor co-agonist, D-serine, in the neurons. This enzyme is also the only bonafide racemase enzyme in human biology at this juncture. As is noted in the manuscript, itis now known that D-serine binds to the "glycinesite" on the NMDAR a full two orders of magnitude more tightly than glycine at the "glycine site,"elevating interest in hSR, the enzyme that produces this important neuromodulator.As such, there is great interest in the chemical biology and neurobiology/neuronal signaling communities in gaining more structural and functional information about hSR.

As communicated nicely in this submission, Benjamin Bax and John Atackand their collaborators have succeeded in solving several illuminating X-ray crystal structures of hSR. The work nicely builds on their recent inaugural contribution to the structural biology of hSR released in 2020as the 6SLHstructure in the PDB and in the following paper:Acta Crystallographica,Section FMS (ref. 32 in the MS and ref.1 in the SI: doi:10.1107/S2053230X920001193).

Here, the team describes both a new"control"open structure of the internal aldimine (6ZUJ,1.80 Å) that is similar to the 6SLHstructure. Also, the authors report the first ATP-bound structure of the human enzyme, a closedstructure that also features a bound malonate(6ZSP, 1.60 Å). The authors note that Y121 is a conserved residue across a range of eukaryotic SR enzymes, from mouse to rat to human to maize and Schizosaccharomyces pombeand that this residue interacts with the active site re-face base, Ser-84, in the resting internal aldimine stateas is clearly seen the 6ZUJstructureof the enzyme whereATP is not bound.They then note, and as was previously seenin 1WTC, the S.pombestructure with AMPPCP bound,thata keytyrosine residue (Y119 there--as can be seen from the alignment presented in SI-Supplementary Figure 11;Y121in the human enzyme)interacts directly with the alpha-phosphate of the bound ATP or analogue. This allows the authors to make the important observation that Y121 movement is a critical component of the

conformational change associated with ATP binding to the SR enzyme at the dimer interface.

In addition to these two valuable crystal structures, the authors also provide herein the first successful demonstration of a serial, crystallographic ligand-soaking screen for this enzyme. And while this done in 30-40% DMSO and @150 mM fragment concentration across the library to bias the system in favor of the binding of relatively hydrophobic fragments (this bias should be explicitly discussed), significant results were obtained, with structures 7NBC-7NBH (all between 1.53 and 1.86 Å) revealing five new types of fragments bound across four new loci:

- Fragment x0430 is bound to a pocket in the small subunit
- Fragment x0478 is bound to a hydrophobic 4W pocket (W275; W325, W275', W325')
- Fragments x0406 & x0495 – are both bound in between the small and large domains
- Fragment x0306 binds at the dimer interface

Interestingly, these results also allow the authors to identify a putative binding pocket for a pseudo-tripeptide ligand previously identified by Toney and coworkers; namely N-phenylpropanoyl-L-His-L-homoPhe-NH₂ to bind where fragment x0495 binds which is actually at original Y121 locus in the open structure! This overlay is quite compelling and shows the value of the ligand bound structures revealed herein.

All in all, this full paper provides a very important set of seven useful high resolution structures of the hSR enzyme, revealing an interesting role for Y121 in the conformational change upon ATP binding, and revealing several new loci where small molecule fragments can bind to the enzyme. These latter results are the first such fragment-soaking experiments to have been successfully conducted with this important enzyme and the resulting five structures will be of great interest to the readership of Nature Communications Biology, particularly to the community of biologists and neuroscientists studying hSR structure/function.

I highly support communicating these results at this time, but also strongly recommend to the following revisions:

(1) Change the title – it is presumptuous to speak of a “druggable pocket” at this stage of the development. We do not even know if hSR will be a “druggable target” yet, let alone speak of it having “druggable pockets”! Suggested alternatives: “Tyrosine-121 moves revealing a potentially ligandable pocket. . .” or “Tyrosine-121 moves opening up a potential binding pocket. . .”

(2) Supplementary Tables 5 and 6, presented in the SI here, summarizing respectively, all reported x-ray crystal structures of the SR enzyme and all known inhibitors of the enzyme are both very useful and should not be buried in the SI. I would like to see these brought out in the body of the paper.

(3) While the authors do provide some comparison of the two ATP (or analogue)-bound eukaryotic SR structures, 1WTC (*S. pombe*-earlier) and 6ZSP (human – here) in the Supporting Information, they should do so more explicitly in the manuscript itself. Please show an overlay of those structures in the body of the paper and describe similarities and differences. As discussed above, the role of Y119 (*S. pombe*) and Y121 (human) in forming a strong H-bond to the alpha-phosphate of the bound AMP/PCP/ATP appears to be conserved. Yet the authors claim that, “Comparison of hSR and spSR holo structures reveals the binding pocket of the ATP analogue in spSR does not exist in the ‘open’ hSR structures.” The authors should clarify this by clearly illustrating which features are conserved and which are distinct in this overlay.

(4) When appearing at the beginning of a sentence, *D-Serine* must be capitalized as such, as the D-stereochemical descriptor does not count as the first letter of the word – This is well accepted and widely understood in the biochemical literature (see Lines 12 and 17, for example – check throughout the MS).

(5) Typo on line 27 and on line 75 – should read “aminoacrylate that can tautomerize to an

iminopyruvate tautomer" (iminopropionate is the typo –this should be iminopropionate but even this seems like a misnomer here as this is an imine derivative of pyruvate).

(6) Change "Saccharomyces pombe" to "Schizosaccharomyces pombe" throughout.

Dr Ben Bax
 Medicines Discovery Institute
 Cardiff University
 Main Building
 Park Place
 Cardiff, CF10 3AT
 29th October 2021

Please find below our answers to referees comments on a manuscript that was originally titled:

"Tyrosine 121 moves revealing a **druggable** pocket that couples catalysis to ATP-binding in serine racemase"

And is now titled:

"Tyrosine 121 moves revealing a **ligandable** pocket that couples catalysis to ATP-binding in serine racemase"

We have followed advice on your website and have prepared a table describing our answers to the referees comment (original comments are at the end of the rebuttal letter).

Specific answers to referee's criticism is given below.

No.	Referees Comment	Our response
Referee 1.		
1.0	The manuscript is thoroughly written, giving a very detailed introduction, also including a comparison to corresponding yeast structures. I have no major objections, only a few minor comments:	We very much appreciate the referee's supportive comments.
1.1	Unless it is done during the editorial process, the manuscript should be carefully proofread to correct for several typos.	Manuscript has now been proof-read and spell checked.
1.2	On page 17, line 340, the authors describe the position of compound 14 above Leu281. This residue should also be labelled in the figure.	Leu 281 now labelled on fig. 6a (see also 1.5).
1.3	Reference 32: The full reference including pages should be given.	Page numbers now included for this reference
1.4	Reference 48 and 55 are the same.	Error corrected
1.5	I do not like some of the figures; however, this is my personal opinion. For example in Fig. 3c, why is the semi-transparent surface shown? This distracts from the intended superposition of the molecules. Similar, what is the purpose of the multi-coloured semi-transparent surface of Fig. 6a and 6c? Here, the important part is the position of the bound fragment, which is hard to recognize in 6a and poorly visible in 6c. Instead a close-up view of the ligands with interacting residues should be shown. Same	The referees point is well made - some panels figures were trying to show both the 'wood' and the 'trees'. In Fig. 3c a semi-transparent surface is shown so the reader can see the 'cleft' between the large and small domains (a 'wood' view). Compound is seen in Fig. 3a. Views of compound binding sites in Fig. 6a and 6c have been enlarged so compounds and residues forming their binding pockets can be clearly seen (- now 'tree' views).

	for Fig. 7a and 7b. The legend of Fig. 7b describes hydrogen bonds to Ser84 and Asp238, which I do not recognize in the figure.	Fig. 7a has been enlarged so compounds can be clearly seen and hydrogen bonds to Ser84 and Asp238 are now clearly shown in Fig. 7a (now a 'tree' view). Legend has been revised to reflect changes in revised figure.
1.6	Supplemental Material: 6. Table S1, second column: Instead of the 2D drawing, I would include the compound number, as the 2D description is also given in Table S2.	Done as suggested. Compounds 1-12 are seen in tyrosine 121 pocket. 13, 14 and 15 are in other sites.
1.7	Table S1: Are the cell dimensions determined with different accuracies? If not, they should have the same number of digits after the period.	Corrected. All cell dimensions now with two digits after the period.
1.8	Table S4: How can the % inhibition be above 100% at 10 mM concentration? Also, is there an explanation why for compound 6, inhibition at 1mM is larger than at 10mM, assuming values above 100% are meaningful in the first place?	We have now measured IC50s on a number of compounds, and replaced the % inhibition data with more informative IC50s in Table S4. Raw data are shown in new supplementary Fig. S12. A full explanation of how values are calculated is now given in Methods.
1.9	References: Check references 10, 23, 30, 50 and 54. I doubt these authors have only single character names. Reference 31 and 41 are the same.	We thank the referee for noting this in the supplementary references. All now checked and full names now given for supplementary references 10, 23, 30, 50 and 54.
Referee 2.		
2.0	Overall, the study has been carried out rigorously, nicely presented and discussed. A long waited structure of the human enzyme in the presence of the allosteric effector ATP has been determined. A few points should be addressed:	We thank referee 2 for a positive response to the manuscript.
2.1	Line 141 Quinoid should read quinonoid	Change made as requested - in main text and supplementary.
2.2	Line 354-356 Authors might rephrase the sentence that it is unclear with too many "closer", "closed".	Sentence rephrased to simplify, now reads: 'Small movements at the dimer interface mean the pockets into which fragments x0478 and x0306 have bound in the 'open' structure do not exist in the 'closed' ATP/malonate structure (Fig. 6b, d).'
2.3	Line 419-422 "It seems unlikely that both subunits in the dimer bind serine and catalyse reactions concurrently. The purpose of the second subunit in the dimer may be to help guide the 'active' subunit in the dimer along a particular catalytic pathway." This is indeed a strong statement as it implies a half of the site reactivity for serine racemase that has never been reported. The same is true for all enzymes belonging to fold type 2 of the	We have rephrased the penultimate paragraph of the discussion - which points to a thorough discussion of mechanism in the SI. 'The evolution and regulation of the activity of human serine racemase in the CNS') and to three movies - and now reads: It is interesting that, although SR functions as a dimer, all the active site residues are located in a single subunit. Serine racemase may have evolved from a serine dehydratase enzyme ¹⁹

	PLP-dependent family. Authors cannot derive only from structural data functional implications (see also below). Unfortunately, this line of thinking is still very popular among protein crystallographers. A structure does not tell us almost nothing on function. Efforts should be devoted to collect structural and functional data, potentially in the same physical state either in the crystalline state via microspectrophotometric studies or in solution via cryo-EM, and correlate them.	(see supplementary discussion and supplementary Fig. 15). Perhaps, for one or two of the four reactions catalysed by SR, the purpose of the second subunit in the dimer may be to help guide the 'active' subunit in the dimer along a particular catalytic pathway. We have made three movies illustrating structural changes between 'open' and 'closed' conformations (Supplementary Movies S1, S2, S3). We note the comments of the referee, but think informed scientific speculation is appropriate at the end of the discussion.
2.4	Line 525. The use of PMP, instead of PLP, seems quite odd and absolutely confusing. In literature and biochemistry textbook PLP is the active form of vitamin B6 that binds to enzyme active sites via a reversible Schiff base (or internal aldimine). PMP stands for the derivative that is formed in the transamination reaction upon transfer of the alpha amine of an amino acid to the internal aldimine. I suggest to retain the term PLP or, otherwise, reports as a footnote, IUPAC indications. If any, of the change in terminology.	We agree that PLP is in common useage in the literature. My error - I confused the free amine name with the Schiff base (or internal aldimine). I have now corrected this to remove all references to the PMP - in the manuscript (including supplementary).
2.5	Authors are proposing a key role for Tyr121 on the basis only of structural observations, i.e. the different conformation of Tyr121 in the absence and presence of ATP and potential inhibitors that displace Tyr121 from its position in the closed conformation. However, a structural observation that not necessarily translates in the activation/inhibition of the enzyme. To prove the point authors should substitute Tyr121 and determine the effect on ATP activation, as it is usually carried out in enzymology, and specifically on serine racemase (Canosa et al.) for assessing the role of Gln79. On this basis, even the paper title is partially misleading as the effective role of Tyr121 in the enzyme activation is not proved.	We have made three mutants of tyrosine 121. Y121F, Y121A and Y121G and assayed them at various ATP concentrations - as shown in new supplementary Fig. 11. The Y121F, Y121A and Y121G mutants have lower activity are less active in 1mM ATP (the concentration used for inhibitor assays). We also assayed the inhibition of the three mutants in the presence of compounds to give IC50s (in supplementary Table 4 - raw data in supplementary Figure 12).
2.6	With the determination of human ATP-bound serine racemase structure authors are in the position to nicely discuss recent studies on the role of Cys113 nitrosylation (Marchesani et al, BBA, 2018; Marchesani et al. Febs Journal, 2020) as well as other	We thank the referee for comments. Three of these papers are now referenced (51, 52 and 54) in the main paper. A paper on enzyme inhibition by NADH derivatives (Bruno et al., 2016) is referenced in the supplementary discussion. This is from the

	previous studies of enzyme inhibition by NADH derivatives (Bruno et al., BBA, 2016) and ATP activation (Marchetti et al., 2013).	Biochemical Journal not BBA - presumably this is the paper the referee is referring to? (Bruno, S. et al. Human serine racemase is allosterically modulated by NADH and reduced nicotinamide derivatives. Biochem J 473 , 3505-3516, doi:10.1042/BCJ20160566 (2016).
2.7.	IC50s for the inhibitors for which structures are reported should be determined. Authors did not try any structure-activity relationship on the list of identified inhibitors. This might be of help for the improvement in affinity.	IC50s are now detailed in supplementary table 4, both for the wild-type and tyrosine 121 mutants. These structure activity relationships will be useful to guide structure-based ligand design.
Referee 3.		
3.0	All in all, this full paper provides a very important set of seven useful high resolution structures of the hSR enzyme, revealing an interesting role for Y121 in the conformational change upon ATP binding, and revealing several new loci where small molecule fragments can bind to the enzyme. These latter results are the first such fragment-soaking experiments to have been successfully conducted with this important enzyme and the resulting five structures will be of great interest to the readership of Nature Communications Biology , particularly to the community of biologists and neuroscientists studying hSR structure/function. I highly support communicating these results at this time, but also strongly recommend to the following revisions:	We thank referee 3 for positive response to the manuscript and strong support for publication.
3.1	Change the title—it is presumptuous to speak of a “druggable pocket” at this stage of the development. We do not even know of hSR will be a “druggable target” yet, let alone speak of it having “druggable pockets”! Suggested alternatives: “Tyrosine-121 moves revealing a potentially ligandable pocket. . .” or “Tyrosine-121 moves opening up a potential binding pocket. . .”	We have changed the title. The word druggable - has been replaced with ligandable ' The ATP-binding residue Tyrosine 121 moves revealing a ligandable pocket in serine racemase '. To date we have clearly seen ligands occupying this pocket. See also 2.5. We now include IC50s for some of the ligands that occupy this pocket.
3.2	Supplementary Tables 5 and 6, presented in the SI here, summarizing respectively, all reported x-ray crystal structures of the SR enzyme and all known inhibitors of the enzyme are both very useful and should not be buried in the SI. I would like to see these brought out in the body of the paper.	We thank the referees for the suggestion. However, due to the limited number of figures/tables available in the main body of the paper we cannot justify moving these two ' very useful ' tables: Supplementary Table 5. List of serine racemase (SR) and serine dehydratase (SDH) structures from the PDB. (Now Supplementary Table 2).

		Supplementary Table 6. Summary of SR inhibitors and physiological modulators Currently these two ' very useful ' tables remain present in the SI.
3.3	While the authors do provide some comparison of the two ATP(or analogue)-bound eukaryotic SR structures, 1 WTC(S. pombe -earlier) and 6ZSP(human –here) in the Supporting Information, they should do so more explicitly in the manuscript itself. Please show an overlay of those structures in the body of the paper and describe similarities and differences. As discussed above, the role of Y119(S. pombe) and Y121 (human) in forming a strong H-bond to the alpha-phosphate of the boundAMPPCP/ATP appears to be conserved. Yet the authors claim that, "Comparison of hSR and spSR holo structures reveals the binding pocket of the ATP analogue in spSR does not exist in the 'open' hSR structures." The authors should clarify this by clearly illustrating which features are conserved and which are distinct in this overlay.	We have made modified Supplementary Fig. 2g, to make this point more clearly. Some papers in the literature have wrongly assumed that yeast and human apo serine racemase structures are the same. This is clearly not true and anyone looking at structures already in the pdb can see this. This is also described in the text in several places for example at the end of section 2.4, which now says: 'Differences between 'open' yeast (1V71 and 1WTC) and 'open' hSR structures (5x2l, 6slh, 6zuj, 7nbc, 7nbd, 7nbf, 7nbg and 7nbh) means the ATP binding pocket does not exist in the 'open' hSR structure (supplementary Fig. 2g).'
3.4	When appearing at the beginning of a sentence, *D-Serine*must be capitalized as such, as the D-stereochemical descriptor does not count as the first letter of the word –This is well accepted and widely understood in the biochemical literature (see Lines 12 and 17, for example –check throughout the MS).	Requested changes made.
3.5	Typo on line 27and on line 75–should read "aminoacrylate that can tautomerize to an iminopyruvate tautomer" (iminopropionate is the typo –this should be iminopropionate but even this seems like a misnomer here as this is an imine derivative of pyruvate).	Typo corrected - to 2-iminopropanoate.
3.6	Change "Saccharomyces pombe" to "Schizosaccharomyces pombe" throughout.	Schizo added.

Referee expertise:

Referee #1: structure based drug discovery, enzymes

Referee #2: X-ray crystallography, protein biochemistry, PLP enzymes

Referee #3: PLP enzymes, serine racemase, XFELs

Reviewers' comments:

Reviewer #1 (Remarks to the Author):

The manuscript by Koulouris et al. describes the investigation of a series of human serine racemase (hSR) structures obtained in a fragment screening campaign. One structure was determined for the first time as the "closed" crystal structure of hSR bound to ATP. In addition of a holo structure of hSR in the "open" conformation, five "open" structures bound to fragments were determined. A key role in the catalytic mechanism is assigned to Tyr121. Two fragments force the sidechain of Tyr121 into an out position and thus suggest a druggable pocket.

The manuscript is thoroughly written, giving a very detailed introduction, also including a comparison to corresponding yeast structures. I have no major objections, only a few minor comments:

1. Unless it is done during the editorial process, the manuscript should be carefully proofread to correct for several typos.
2. On page 17, line 340, the authors describe the position of compound 14 above Leu281. This residue should also be labelled in the figure.
3. Reference 32: The full reference including pages should be given.
4. Reference 48 and 55 are the same.
5. I do not like some of the figures; however, this is my personal opinion. For example in Fig. 3c, why is the semi-transparent surface shown? This distracts from the intended superposition of the molecules. Similar, what is the purpose of the multi-coloured semi-transparent surface of Fig. 6a and 6c? Here, the important part is the position of the bound fragment, which is hard to recognize in 6a and poorly visible in 6c. Instead a close-up view of the ligands with interacting residues should be shown. Same for Fig. 7a and 7b. The legend of Fig. 7b describes hydrogen bonds to Ser84 and Asp238, which I do not recognize in the figure.

Supplemental Material:

6. Table S1, second column: Instead of the 2D drawing, I would include the compound number, as the 2D description is also given in Table S2.
7. Table S1: Are the cell dimensions determined with different accuracies? If not, they should have the same number of digits after the period.
8. Table S4: How can the % inhibition be above 100% at 10 mM concentration? Also, is there an explanation why for compound 6, inhibition at 1mM is larger than at 10mM, assuming values above 100% are meaningful in the first place?
9. References: Check references 10, 23, 30, 50 and 54. I doubt these authors have only single character names. Reference 31 and 41 are the same.

Reviewer #2 (Remarks to the Author):

The manuscript by Koulouris et al. reports on the investigation of the PLP-dependent enzyme serine racemase exploiting both structural and kinetic methods. Results led authors to propose a key role for Tyr121 in the enzyme activation by coupling ATP site with the active site. In addition, authors have identified a novel class of enzyme inhibitors acting by displacing Tyr121 and stabilizing an open, inactive enzyme conformation.

Overall, the study has been carried out rigorously, nicely presented and discussed. A long waited

structure of the human enzyme in the presence of the allosteric effector ATP has been determined. A few points should be addressed:

Line 141 . Quinoid should read quinonoid.

Line 354-356 Authors might rephrase the sentence that it is unclear with too many "closer", "closed".

Line 419-422 "It seems unlikely that both subunits in the dimer bind serine and catalyse reactions concurrently. The purpose of the second subunit in the dimer may be to help guide the 'active' subunit in the dimer along a particular catalytic pathway." This is indeed a strong statement as it implies a half of the site reactivity for serine racemase that has never been reported. The same is true for all enzymes belonging to fold type 2 of the PLP-dependent family. Authors cannot derive only from structural data functional implications (see also below).

Unfortunately, this line of thinking is still very popular among protein crystallographers. A structure does not tell us almost nothing on function. Efforts should be devoted to collect structural and functional data, potentially in the same physical state either in the crystalline state via microspectrophotometric studies or in solution via cryo-EM, and correlate them.

Line 525. The use of PMP, instead of PLP, seems quite odd and absolutely confusing. In literature and biochemistry textbook PLP is the active form of vitamin B6 that binds to enzyme active sites via a reversible Schiff base (or internal aldimine). PMP stands for the derivative that is formed in the transamination reaction upon transfer of the alpha amine of an amino acid to the internal aldimine. I suggest to retain the term PLP or, otherwise, reports as a footnote, IUPAC indications. If any, of the change in terminology.

Authors are proposing a key role for Tyr121 on the basis only of structural observations, i.e. the different conformation of Tyr121 in the absence and presence of ATP and potential inhibitors that displace Tyr121 from its position in the closed conformation. However, a structural observation that not necessarily translates in the activation/inhibition of the enzyme. To prove the point authors should substitute Tyr121 and determine the effect on ATP activation, as it is usually carried out in enzymology, and specifically on serine racemase (Canosa et al.) for assessing the role of Gln79. On this basis, even the paper title is partially misleading as the effective role of Tyr121 in the enzyme activation is not proved.

With the determination of human ATP-bound serine racemase structure authors are in the position to nicely discuss recent studies on the role of Cys113 nitrosylation (Marchesani et al, BBA, 2018; Marchesani et al. Febs Journal, 2020) as well as other previous studies of enzyme inhibition by NADH derivatives (Bruno et al., BBA, 2016) and ATP activation (Marchetti et al., 2013).

IC50s for the inhibitors for which structures are reported should be determined.

Authors did not try any structure-activity relationship on the list of identified inhibitors. This might be of help for the improvement in affinity.

Reviewer #3 (Remarks to the Author):

Human serine racemase(hSR)is the enzyme responsible forthe biosynthesis of the important NMDA receptor co-agonist, D-serine, in the neurons. This enzyme is also the only bonafide racemase enzyme in human biology at this juncture. As is noted in the manuscript, itis now known that D-serine binds to the "glycinesite" on the NMDAR a full two orders of magnitude more tightly than glycine at the "glycine site,"elevating interest in hSR, the enzyme that produces this important neuromodulator.As such, there is great interest in the chemical biology and neurobiology/neuronal signaling communities in gaining more structural and functional information about hSR.

As communicated nicely in this submission, Benjamin Bax and John Atackand their collaborators have succeeded in solving several illuminating X-ray crystal structures of hSR. The work nicely builds on

their recent inaugural contribution to the structural biology of hSR released in 2020 as the 6SLH structure in the PDB and in the following paper: *Acta Crystallographica, Section F* (ref. 32 in the MS and ref. 1 in the SI: doi:10.1107/S2053230X920001193).

Here, the team describes both a new "control" open structure of the internal aldimine (6ZUJ, 1.80 Å) that is similar to the 6SLH structure. Also, the authors report the first ATP-bound structure of the human enzyme, a closed structure that also features a bound malonate (6ZSP, 1.60 Å). The authors note that Y121 is a conserved residue across a range of eukaryotic SR enzymes, from mouse to rat to human to maize and *Schizosaccharomyces pombe* and that this residue interacts with the active site re-face base, Ser-84, in the resting internal aldimine state as is clearly seen in the 6ZUJ structure of the enzyme where ATP is not bound. They then note, and as was previously seen in 1WTC, the *S. pombe* structure with AMPPCP bound, that a key tyrosine residue (Y119 there--as can be seen from the alignment presented in SI-Supplementary Figure 11; Y121 in the human enzyme) interacts directly with the alpha-phosphate of the bound ATP or analogue. This allows the authors to make the important observation that Y121 movement is a critical component of the conformational change associated with ATP binding to the SR enzyme at the dimer interface.

In addition to these two valuable crystal structures, the authors also provide herein the first successful demonstration of a serial, crystallographic ligand-soaking screen for this enzyme. And while this was done in 30-40% DMSO and @150 mM fragment concentration across the library to bias the system in favor of the binding of relatively hydrophobic fragments (this bias should be explicitly discussed), significant results were obtained, with structures 7NBC-7NBH (all between 1.53 and 1.86 Å) revealing five new types of fragments bound across four new loci:

- Fragment x0430 is bound to a pocket in the small subunit
- Fragment x0478 is bound to a hydrophobic 4W pocket (W275; W325, W275', W325')
- Fragments x0406 & x0495 are both bound in between the small and large domains
- Fragment x0306 binds at the dimer interface

Interestingly, these results also allow the authors to identify a putative binding pocket for a pseudo-tripeptide ligand previously identified by Toney and coworkers; namely N-phenylpropanoyl-L-His-L-homoPhe-NH₂ to bind where fragment x0495 binds which is actually at original Y121 locus in the open structure! This overlay is quite compelling and shows the value of the ligand bound structures revealed herein.

All in all, this full paper provides a very important set of seven useful high resolution structures of the hSR enzyme, revealing an interesting role for Y121 in the conformational change upon ATP binding, and revealing several new loci where small molecule fragments can bind to the enzyme. These latter results are the first such fragment-soaking experiments to have been successfully conducted with this important enzyme and the resulting five structures will be of great interest to the readership of *Nature Communications Biology*, particularly to the community of biologists and neuroscientists studying hSR structure/function.

I highly support communicating these results at this time, but also strongly recommend to the following revisions:

(1) Change the title--it is presumptuous to speak of a "druggable pocket" at this stage of the development. We do not even know if hSR will be a "druggable target" yet, let alone speak of it having "druggable pockets"! Suggested alternatives: "Tyrosine-121 moves revealing a potentially ligandable pocket. . ." or "Tyrosine-121 moves opening up a potential binding pocket. . ."

(2) Supplementary Tables 5 and 6, presented in the SI here, summarizing respectively, all reported x-ray crystal structures of the SR enzyme and all known inhibitors of the enzyme are both very useful and should not be buried in the SI. I would like to see these brought out in the body of the paper.

(3) While the authors do provide some comparison of the two ATP(or analogue)-bound eukaryotic SR structures, 1 WTC(*S. pombe*-earlier) and 6ZSP(human –here) in the Supporting Information, they should do so more explicitly in the manuscript itself. Please show an overlay of those structures in the body of the paper and describe similarities and differences. As discussed above, the role of Y119(*S. pombe*) and Y121 (human) in forming a strong H-bond to the alpha-phosphate of the bound AMPPCP/ATP appears to be conserved. Yet the authors claim that, "Comparison of hSR and spSR holo structures reveals the binding pocket of the ATP analogue in spSR does not exist in the 'open' hSR structures." The authors should clarify this by clearly illustrating which features are conserved and which are distinct in this overlay.

(4) When appearing at the beginning of a sentence, *D-Serine* must be capitalized as such, as the D-stereochemical descriptor does not count as the first letter of the word –This is well accepted and widely understood in the biochemical literature (see Lines 12 and 17, for example –check throughout the MS).

(5) Typo on line 27 and on line 75 –should read "aminoacrylate that can tautomerize to an iminopyruvate tautomer" (iminopropionate is the typo –this should be iminopropionate but even this seems like a misnomer here as this is an imine derivative of pyruvate).

(6) Change "Saccharomyces pombe" to "Schizosaccharomyces pombe" throughout.

Reviewers' comments:

Reviewer #1 (Remarks to the Author):

As I only had minor comments in the previous review, I do not object publication. Most of my points were addressed.
References 31 and 41 in the Supplemental Material are still the same. This should be corrected.

One question out of curiosity. In the originally submitted manuscript, for compound x0495, 102% inhibition was given at 10mM.

In the revised Table S4 following is stated "Note only 3 compounds, 3-x0458, 4-x0482 and 13-x0430 showed 100% inhibition". What caused this change in inhibition for compound x0495?

Reviewer #2 (Remarks to the Author):

Authors have properly addressed most of points. However, they missed to modify the term quinoid in quinonoid in Fig. 1 and 9.

Authors should better explain the activity assay. Which is the product that is associated with emission of light?

Was the ATP concentration checked during the crystallization trials? ATP is quite unstable releasing phosphate.

I am still very uncomfortable with the statement that in SR only one subunit is active. This statement is not supported by any experimental data and should be deleted. It is not a speculation, it is a misleading sentence. Each subunit binds a PLP molecules and each subunit possesses the catalytic residues. Authors might carry out experiments to support their view, such as titration with malonate as a function of activity and parallel x-ray crystallography, or titration of the apo-enzyme with PLP vs activity, i.e. how many either malonate or PLP molecules are needed to abolish SR activity?

Reviewer #3 (Remarks to the Author):

This submission from Bax and coworkers has been significantly improved from the original version. Notably the authors have carefully kinetically examined the fragments that were shown to bind to the hSR enzyme through crystal-soaking experiments, and report IC(50) values for each of these. This provides a nice activity-based complementary data set to the structural biological data sets presented as the primary data from this study, data sets that the community will find very interesting I am sure!

The fact that this work now allows the authors to postulate a binding site for the pseudo-tripeptide inhibitor identified early on by Toney and Kurth and coworkers is particularly valuable.

The authors have also now conducted helpful site-directed mutagenesis studies on hSR having made three mutants; namely Y121F, Y121A and Y121G and assayed the activity of these mutants at various ATP concentrations (New SI Fig. 11). The fact that the Y121F, Y121A and Y121G mutants have lower catalytic activity than the wt-hSR in the presence of 1 mM ATP is consistent with the hypothesis put forward that the Y121 is critical for allosteric activation of the enzyme by ATP.

Corrections/Additions:

Page 3: Change "catalyzing the beta-elimination of the side chain of hydroxyl of serine . . . ," to "catalyzing the beta-elimination of the elements of water from L-serine, thereby producing ammonia and pyruvate, following enamine tautomerization and hydrolysis."

Note: I would recommend replacing the discussion about tautomers, etc. with this text, as the current description implies release of tautomeric products into solution, which may well not be the

case!

In the Introduction, authors should consider making reference to the beautiful work of Coyle and coworkers that suggests that neurotoxic reactive astrocytes express elevated levels of human serine racemase in Alzheimer's Disease:

<https://pubmed.ncbi.nlm.nih.gov/31212068/>

Pursuant to these changes, I would recommend publication of this manuscript in Nature Bio Comms.

14th January 2022

Re: Koulouris et al: Tyrosine 121 moves revealing a ligandable pocket that couples catalysis to ATP-binding in serine racemase

Dear Dr. Kumar,

We thank you for your E-mail of 16th November 2021 and apologies for our delayed response only a number of end-of-year deadlines got in the way. We have addressed the points raised by the referees as listed below and the revised manuscript has been resubmitted using the link you provided.

Summary of changes made

Reviewer #1: All changes made as suggested.

Reviewer #2: Changes made as suggested. As regards speculation at the end of the discussion, Reviewers #1 and #3 do not object so we think that this should be an editorial decision.

Reviewer #3: All changes made as suggested.

Details of changes made

Reviewer #1 (Remarks to the Author):

As I only had minor comments in the previous review, I do not object publication. Most of my points were addressed.

- 1. References 31 and 41 in the Supplemental Material are still the same. This should be corrected.*
This duplicate reference in the Supplemental Material has been removed.
- 2. One question out of curiosity. In the originally submitted manuscript, for compound x0495, 102% inhibition was given at 10mM.*
In the revised Table S4 following is stated "Note only 3 compounds, 3-x0458, 4-x0482 and 13-x0430 showed 100% inhibition". What caused this change in inhibition for compound x0495?
The change in inhibition, is probably just down to experimental errors on original data (only two data points).

Reviewer #2 (Remarks to the Author):

- 1. Authors have properly addressed most of points. However, they missed to modify the term quinoid in quinonoid in Fig. 1 and 9.*
Modifications now made as requested.
- 2. Authors should better explain the activity assay.*
Modifications now made as requested (first paragraph - section 4.6).
- 3. Which is the product that is associated with emission of light?*
Modifications now made as requested (first paragraph - section 4.6).
- 4. Was the ATP concentration checked during the crystallization trials?*
No - but the electron density maps are very clear - we were surprised to observe the ATP
- 5. I am still very uncomfortable with the statement that in SR only one subunit is active.*

We have rephrased this sentence to read: "***It is possible that, for one or two of the four reactions catalysed by SR, the purpose of the second subunit in the dimer may be to help guide the 'active' subunit in the dimer along a particular catalytic pathway.***" It is in the penultimate paragraph of the discussion. We think a little speculation in the discussion is reasonable and the sentence starts with the phrase '***It is possible that***'. However, if the editor insists on removing the sentence we will not object.

6. *This statement [that in SR only one subunit is active] is not supported by any experimental data and should be deleted. It is not a speculation, it is a misleading sentence. Each subunit binds a PLP molecules and each subunit possesses the catalytic residues. Authors might carry out experiments to support their view, such as titration with malonate as a function of activity and parallel x-ray crystallography, or titration of the apo-enzyme with PLP vs activity, i.e. how many either malonate or PLP molecules are needed to abolish SR activity?*

We thank Reviewer #2 for the suggestions for additional experiments, which we agree are an excellent way of addressing our hypothesis. Unfortunately, however, funding for this project has ended and the very talented early career researcher that conducted the majority of these studies (Dr. Koulouris) has moved to another position.

Reviewer #3 (Remarks to the Author):

1. *Page 3: Change "catalyzing the beta-elimination of the side chain of hydroxyl of serine . . . , ' to "catalyzing the beta-elimination of the elements of water from L-serine, thereby producing ammonia and pyruvate, following enamine tautomerization and hydrolysis."*

Note: I would recommend replacing the discussion about tautomers, etc. with this text, as the current description implies release of tautomeric products into solution, which may well not be the case!

Modification made as suggested, to: 'SR can also act as a dehydratase, catalyzing the β -elimination of the elements of water from L-serine, thereby producing ammonia and pyruvate, following enamine tautomerization and hydrolysis'.

2. *In the Introduction, authors should consider making reference to the beautiful work of Coyle and coworkers that suggests that neurotoxic reactive astrocytes express elevated levels of human serine racemase in Alzheimer's Disease: <https://pubmed.ncbi.nlm.nih.gov/31212068/>*

Thank you for bringing this to our attention. This work is now explicitly described in the Introduction.

Pursuant to these changes, I would recommend publication of this manuscript in Nature Bio Comms.

We hope that these changes we have made are to your satisfaction.

Yours,

Prof. John R. Atack

Dr. Ben Bax

REVIEWERS' COMMENTS:

Reviewer #2 (Remarks to the Author):

Authors have properly addressed my comments.